

# On the meridional ageostrophic transport in the tropical Atlantic

Yao Fu[1], Johannes Karstensen[1], Peter Brandt[1]

[1]GEOMAR Helmholtz Centre for Ocean Research Kiel, Kiel, Germany

*Correspondence to*: Yao Fu (yfu@geomar.de)

**Abstract.** The meridional Ekman volume, heat and salt transport across two trans-Atlantic sections near 14.5° N and 11° S were estimated using wind products, in-situ observations, and model data. A meridional ageostrophic velocity was obtained as the difference between the directly measured total velocity and the geostrophic velocity derived from observations. Wave-like structures exist in the ageostrophic velocity with 60-80 m vertical scale and large horizontal coherence, which are likely

associated with near-inertial waves. The meridional Ekman transport estimated by integrating the ageostrophic velocity was 6.2 ± 2.3 Sv northward at 14.5° N and 11.7 ± 2.1 Sv southward at 11° S, which agrees well with the predictions from in-situ wind stress data of 6.7 ± 3.5 Sv at 14.5° N and 13.6 ± 3.3 Sv at 11° S. The top of the pycnocline well represents the penetration depth of the Ekman currents at both sections, which was typically 20 m deeper than the local mixed layer depth. We observed that in

the meridional Ekman heat and salt flux calculation, using only the sea surface temperature and salinity data had a negligible impact on the resulting fluxes compared to using temperature and salinity profile data covering the Ekman layer. The errors in the Ekman heat and salt flux calculation were dominated by the uncertainty in the Ekman volume transport estimates.

## 1 Introduction

In the tropical Atlantic Ocean, strong and steady easterly trade winds generate a poleward meridional flow in the surface layer. According to the classical linear theory of Ekman (1905), under the momentum balance between steady wind stress and Coriolis force, the wind-driven flow spirals clockwise with depth, the Ekman spiral, while the vertical integration of the spiral results in a net volume transport to the right of the wind direction (northern hemisphere), the Ekman transport. A

convergence is created in the subtropics, where the poleward Ekman transport induced by the trade





winds interacts with the equatorward Ekman transport induced by the mid-latitude Westerlies. In simple linear vorticity theory, the Ekman convergence in subtropics drives an equatorward Sverdrup transport that explains many aspects of the wind driven gyre circulation, such as Subtropical Cell (STC). Schott et al. (2004) calculated the Ekman divergence (21-24 Sv, 1 Sv = $10^6$ m$^3$ s$^{-1}$) between 10° N and 10° S in

the tropical Atlantic using climatological wind to infer the strength of the STC; Rabe et al. (2008) further analyzed the variability of the STC using the same sections based on assimilation products, and found that on timescales longer than 5 years to decadal, the variability of poleward Ekman divergence leads the variability of geostrophic convergence in the thermocline.

The meridional Ekman transport is, depending on the latitude, an important upper layer contribution

when estimating the strength of the Meridional Overturning Circulation (MOC, Friedrichs and Hall, 1993; Klein et al., 1995; Wijffels et al., 1996). The variations in the meridional Ekman transport has been found to cause barotropic adjustment of the MOC in the ocean interior on different time scales. Cunningham et al. (2007) reported that the upper ocean had an immediate response to the changes in Ekman transport at sub-seasonal to seasonal timescales, while Kanzow et al. (2010) found that on the

seasonal timescale, the Ekman transport was less important than the mid-ocean geostrophic transport, whose seasonal variation was dominated by the seasonal cycle of the wind stress curl. McCarthy et al. (2012) analysed a low MOC case during 2009 and 2010, they also pointed out that on the interannual timescale, although the Ekman transport played a role, its variability was relatively small compared to the variability in mid-ocean geostrophic transport, especially in the upper 1100 m.

Of interest for large scale overturning studies is also the meridional Ekman driven heat and freshwater fluxes that provide an important upper layer constraint, for example, for geostrophic end point arrays (McCarthy et al., 2015; McDonagh et al., 2015). In many cases, sea surface temperature (SST) has been found to be a sufficient constraint for the Ekman layer temperature (Wijffels et al., 1994; the June 1998 case in Chereskin et al., 2002). This probably is not too much of a surprise as the heat flux is primarily

determined by the transport and less by the relative small variability in temperature. However, the unresolved vertical structure of the water column could lead to an unknown bias, for example, due to the difference between the mixed layer depth (MLD) and the depth of Ekman layer. An extreme case has been reported for the Northern Indian Ocean at 8° N at the end of a summer monsoon event



(Chereskin et al., 2002), where the direct Ekman temperature transport was 5% smaller when using the temperature within the top of the pycnocline (TTP, as a proxy of the Ekman layer depth) than using the SST, and the corresponding mean temperature in the Ekman layer is 1.1 °C cooler than the averaged SST. In this case, the mean TTP depth is 92 m deeper than the mean MLD.

Assuming that the meridional upper layer ageostrophic flow is in Ekman balance, the meridional Ekman transport can be estimated indirectly from zonal wind stress data, and directly from observed ageostrophic velocity:

$$merid. Ekman\ Trasnport = \frac{1}{\rho}\frac{\tau_x}{f} = -\int_{-D_E}^{0} v_{Ekman}\,,\qquad(1)$$

where $\tau_x$ is the zonal wind stress, $\rho$ is the density of sea water, $f$ is the Coriolis parameter of the
respective latitude, $v_{Ekman}$ is the ageostrophic Ekman velocity, $D_E$ is the Ekman depth. $D_E$ can be defined as the e-folding scale depth of the Ekman spiral, leading to an analytical solution of $D_E = \sqrt{\frac{2A_v}{f}}$, where $A_v$ is a constant vertical eddy viscosity (Price et al., 1987). Ekman's solution also reveals a surface Ekman velocity $V_0 = \frac{\tau}{\sqrt{\rho^2 f A_v}}$, which is 45° to the right (left) of the wind blowing direction in the northern (southern) hemisphere.

Meridional ageostrophic currents can be calculated from the difference between direct velocity observations ($v_{obs}$) and geostrophic currents ($v_{geos}$). However, when an ageostrophic meridional component that is not in Ekman balance (e.g. inertial currents) ($v_{ageos_{non\ Ekman}}$) is taken into account:

$$v_{Ekman} = v_{obs} - v_{geos} - v_{ageos_{non\ Ekman}},\qquad(2)$$

often $v_{ageos_{non\ Ekman}}$ is assumed to be 0, so $v_{Ekman}$ equals $v_{ageos}$. Direct velocity profile data, for
example from ADCP, and geostrophic velocities, from hydrographic data, are used in such studies (Chereskin and Roemmich, 1991; Wijffels et al., 1994; Garzoli and Molinari, 2001). The Ekman transport is then derived from vertical integration of the $v_{ageos}$.

For both equations it is relevant to recall that the Ekman balance is derived for an ocean with constant vertical viscosity and infinite depth, forced by a steady wind field (Ekman 1905). Such conditions are



not found in the real ocean; therefore, the application of the indirect (eq .1) and direct (eq. 2) approach suffers from different kinds of errors. For the indirect approach (eq. 1) the temporally varying wind field, the momentum flux calculated from the wind speed, and the unknown partitioning of the wind energy input into the Ekman layer at different frequency bands, are probably the most important sources of errors introduced into any Ekman current/transport estimate. For the direct approach, unknown lower integration depth, momentum flux variability, errors introduced by the experimental design (e.g. an shipboard ADCP does not resolve the upper 10-20 m of the flow, which is often assumed equal to the values at the first valid bin) or instrument errors can impact results of the direct approach. But it may have the potential to separate the ageostrophic current into an Ekman and a non-Ekman component (e.g. near-inertial internal waves).

Many observational studies on Ekman dynamics that compare indirect and direct approaches have been conducted in the trade wind regions, where at least the wind stress forcing is relatively constant. Using shipboard ADCP data with Conductivity-Temperature-Depth (CTD) profile data, Chereskin and Roemmich (1991) directly estimated an Ekman transport of $9.3 \pm 5.5\ Sv$ at 11° N in the Atlantic by integrating an ageostrophic velocity from the surface to a depth equivalent to TTP. The ageostrophic velocity was obtained by subtracting the geostrophic velocity from the ADCP velocity. Using a similar direct method, Wijffels et al. (1994) estimated an ageostrophic transport of $50.8 \pm 10\ Sv$ at 10° N in the Pacific. Chereskin et al. (1997) found Ekman transports of $-17.6 \pm 2.4\ Sv$ and $-7.9 \pm 2.7\ Sv$ during and after a southwest monsoon event at 8.5° N in the Indian Ocean, respectively. In all the above studies, the direct estimates agree within 10-20% of the estimates by the using the in-situ wind data (Eq. 1). Both the direct approach and indirect approach also show consistent transport structure across all the basins, which can be seen from the cumulative meridional Ekman transport curves from one boundary to the other. An indication for the existence of an Ekman balance in the upper ocean is evidence of an Ekman spiral. In all the above publications an "Ekman spiral"-like feature has been identified. Because $v_{geos}$ can be estimated only perpendicular to the CTD stations and all studies are based on more or less zonal CTD section, the three dimensional structure of the Ekman spiral can not be obtained. However, the Ekman flow becomes evident by a near-surface maximum of the meridional ageostrophic velocity decreasing smoothly below within the upper 50 to 100 m to zero.



Despite the fact that the zonal wind in the above studies was predominantly uniform in one direction, their ageostrophic velocity alternated north- and southward. The section-averaged ageostrophic velocity profiles often exhibit structures that are clearly non-Ekman. Chereskin and Roemmich (1991) reported a propagation signal of a strong internal wave that was responsible for a peak in their section-integrated

ageostrophic transport profile below the Ekman layer. Garzoli and Molinari (2001) also reported on vertically alternating structures in the section-averaged ageostrophic velocity profile at 6° N in the Atlantic. They proposed several possible candidates that could contribute to creating this structure, such as inertial currents within the latitude range of the North Equatorial Counter Current (NECC), and tropical instability waves with northward and southward velocities. Besides, they argued that the

advective terms in the momentum equations might also produce a large non-Ekman ageostrophic transport in the presence of large horizontal shears between NECC and the northern branch of the South Equatorial Current (nSEC).

The appearance of these non-Ekman ageostrophic structures is not surprising, since it has been long recognized that the temporal variability of the wind field leads to wind energy input into the Ekman

layer at subinertial and near-inertial frequencies. Wang and Huang (2004) estimated the global wind energy input into the Ekman layer at subinertial frequencies (frequency lower than 0.5 cycle per day) to be 2.4 TW, while Watanabe and Hibiya (2002) and Alford (2003) estimated that at near-inertial frequencies the wind energy input was 0.7 and 0.5 TW, respectively. Elipot and Gille (2009) estimated the wind energy input into the Ekman layer for the frequency range between 0 and 2 cpd at 41° S in the

Southern Ocean using surface drifter data. They found that the near-inertial input (between 0.5 f and 2 cpd) contributes to 8% of the total wind energy input (here the "total" means frequency range between 0 and 2 cpd), which may still underestimate the near-inertial contribution due to details in their data. All these studies suggest that at least about 10% of the wind energy (frequency range between 0 and 2 cpd) into the Ekman layer is at near-inertial frequencies, which is used to supply the non-Ekman

ageostrophic motions (inertial oscillation, near-inertial internal waves, etc.). Therefore, complicated structures in the direct observed ageostrophic velocity as reported by Chereskin and Roemmich (1991) and Garzoli and Molinari (2001) can be anticipated.



The purpose of the present study is to estimate the Ekman volume, heat and freshwater transport across two trans-Atlantic sections nominally along 14.5° N and 11° S by using direct and indirect methods to analyse the vertical structure of the ageostrophic flow by using high resolution velocity and hydrographic data. Typically, on a large-scale cruise survey, CTD profile data have station spacing of approximately 30 to 50 nm. High-resolution CTD data, such as XCTD or the recently introduced underway-CTD (uCTD), allow profiling with denser station spacing of about 8 to 10 nm or less. We will use uCTD data to test the sensitivity of the Ekman transport estimates as well as of the heat and salt fluxes with respect to the CTD profile resolution. In order to integrate the two observation-based Ekman transport estimates into the large-scale tropical Atlantic context, we compared our results with the GECCO2 ocean syntheses data. This work is structured as follows. The processing of the data is described in section 2. The vertical and horizontal structure of the ageostrophic velocity, together with the Ekman volume, heat and salt transport estimated using different datasets and different methods are presented and discussed in section 3, followed by a summary in section 4.

## 2 Data

Two trans-Atlantic zonal sections near 14.5° N and 11° S were occupied by the R/V Meteor on three cruise legs from 28 April through 22 July 2013. The 14.5° N section began with cruise M96 off the coast of Trinidad and Tobago, jointly completed by cruise M97 from 20° W to the African coast (Fig. 1). During these surveys, 64 CTD stations were conducted along the 14.5° N section with an average spacing of 40nm (75 km). The section ended on M96 at about 20° W and was continued to the African coast during M97 20 days later. Parallel to the CTD system, the uCTD system was operated between the adjacent CTD stations when the ship was steaming at 10 to 12 kn. In total, 317 uCTD profiles were achieved, with an average spacing of 8nm (15 km). The 11° S section was surveyed during R/V Meteor M98. In this section, the standard CTD was only operated on the shelf and at the shelf break; during the transit across the Atlantic, only the uCTD was in use. All together, 290 uCTD profiles were taken in this study with an average spacing of 11 nm (20 km). Shipboard ADCP and anemometer were in continuous operation through the entire cruises.



### 2.1 CTD and uCTD measurement

The CTD work was carried out with a Sea-Bird Electronic (SBE) 9 plus CTD system. The two temperature sensors were calibrated at the manufacturer just before the cruise M96 in March 2013. The conductivity measurements were calibrated by comparing the bottle stop data with Salinometer

measurements on bottle samples. All CTD system quality control procedures followed the GO-SHIP recommendations (Hood et al., 2010). The accuracy of the CTD data was estimated to be of ±0.001 °C for temperature and ±0.002 g kg$^{-1}$ for salinity.

The uCTD system used at both zonal sections was an Oceanscience Series II UnderwayCTD. It consisted of a probe, a tail and a winch. The probe is equipped with a temperature (SBE-3F), a

conductivity (SBE-4) and a pressure sensor from SBE. A tail spool reloading system allows the rope spooled on the tail to be paid out when the probe falls freely. The sensors record data at a frequency of 16 Hz. For most of the profiles about 250-300 m of rope were spooled on the tail spool (which set the fall depth) and the recording time length was set to 100 seconds and about 1600 data recordings per cast were obtained. From the tail spool the probe sink freely with a nominal speed of 4 m s$^{-1}$. However, due

to the back and forth unspooling of the rope from one end of the tail to the other, the sinking speed typically varies from 3 to 4.5 m s$^{-1}$. After the rope on the tail is paid out completely, the probe still sinks at speed less than 2 m s$^{-1}$ in the last tens of meters of its sinking before winched back to the ship and recovered back to deck. Three probes were used during the two section surveys (#70200126 and #70200068 along the 14.5° N section; #70200068 and #70200138 along the 11° S section). The uCTD

winches were out of service several times during the three cruise legs. Although they were repaired on-board, several measurement gaps were left, for example, between 29° W and 27° W (Fig. 1).

The post-calibration of the uCTD data was done in two major steps, the first step is a sensor calibration procedure, which corrects the temperature sensor error due to viscous heating, the conductivity sensor error due to thermal mass delay, and the lag between the conductivity and temperature sensors; the

second step is data validation in reference to CTD profile data and to thermosalinograph (TSG) data. The first step was done following Ullman and Hebert (2014) (hereafter UH2014), we will briefly describe the process here, for details please refer to their work. The uCTD is an unpumped CTD system, the rapid sinking speed of 4 m s$^{-1}$ allows water passing through the sensor package at 3.56 m s$^{-1}$



(UH2014). This flow rate is much higher than a pumped CTD system (1 m s⁻¹), which leads to a clear viscous heating effect of the uCTD temperature sensor. This was corrected using a steady-state result of Larson and Pedersen (1996) for the perpendicular flow case (cf. Eq. 8 of UH2014). The thermal mass correction was performed following the algorithm of Lueck and Picklo (1990) and using the mean

values of error magnitude and time constant from UH2014 (cf. Table 1 of UH2014).

From the uCTD profiles alone a time lag correction was determined from cross-correlation of temperature and conductivity sensor small-scale variability. The variability was calculated by subtracting a 6ᵗʰ order Butterworth low-pass filtered profile with a cut-off frequency of 4 Hz from the corresponding temperature and conductivity time series of each profile. The highest correlation was

found for a 1/16 s lag (conductivity leading) and which equals the sampling frequency of 16 Hz data. Application of the lag eliminated most of the spikes in the salinity profiles when the sinking speed of the probe was above about 1.5 m s⁻¹. However, when the sinking speed was below 1.5 m s⁻¹, this correction would cause the spikes pointing at the opposite direction and indicates an inverse dependency of the lag on the sinking speed. This result is consistent with that reported by UH2014, and

we corrected the lag following their lag model (cf. Eq. 7 of UH2014), but adjusted their parameters to match our data. The data recorded with a sinking speed smaller than 0.3 m s⁻¹ was neglected (including all upcast data).

Validation of the lag corrected uCTD against CTD profile data revealed for the 14.5° N section a drift in the conductivity sensors of the uCTD probes #70200126 and #70200068. A bias correction in the sense

of a absolute salinity offset (uCTD – CTD) was determined based on the temperature-salinity space (Rudnick and Klinke, 2007) by considering the conservative temperature range from 12° to 14° C and using all uCTDs between adjacent CTD pairs. This particular temperature range was chosen because it belongs to the Atlantic central water, whose T/S relation is nearly linear, which implies that in this temperature range, the spreading of salinity measured during different uCTD casts should be tight.

Besides, it was also surveyed by almost all uCTD casts along the section. For probe #70200126, the salinity offset fluctuates around a mean value of 0.038 g kg⁻¹ west of 39° W (CTD station 34), east of which the offset shifts abruptly to around 0.151 g kg⁻¹. The calibration was done by applying the mean offset values to the salinity data in the corresponding groups of uCTDs. The salinity data of the last few





profiles of probe #70200126 (between 30° W and 29° W) were extremely noisy, and not possible to calibrate. This probe was not further used during the rest of the section due to its poor quality of the salinity data. For probe #70200068, the salinity offset remains around 0 west of 36° W (CTD station 38), and then abruptly shifts to around 0.295 g kg$^{-1}$ between 36° W and 23.5° W (CTD station 56). East of 23.5° W to the African coast, the offset shows a linear decreasing trend. This is likely due to the increasing portion of South Atlantic central water (SACW) in the central water layer when approaching to the coastal region, which is less saline than the North Atlantic central water (NACW), and consequently shifting the slope of the T/S curve. As a result, the linear trend of the offset east of 23.5° W should not be due to instrument error. Therefore, only a mean offset was calculated and applied to calibrate each corresponding group of profiles made by #70200068. The reasons for the abrupt drift in the salinity (as obtained from the conductivity sensors) are not clear, but it is likely that due to the repeated intensive usage, the conductivity sensors were contaminated or impacted (hit ship hull).

The shipboard TSG provides another source of validation and calibration of uCTD data. On R/V Meteor, the TSG (SBE38 for temperature sensor, SBE21 for conductivity sensor) measures temperature and salinity at an intake at approximately 6.5 m depth. For all the three legs, the TSG conductivity cell was calibrated from salinity analysis of water samples taken at the water intake, and a comparison with CTD data (if available) was also done. The uCTD salinity calibration was done by calculating the conductivity offset between the uCTD at 6.5 m and the averaged TSG conductivity within 5-min before and after the uCTD downcast. For probe #70200126, the drift of its conductivity sensor manifests also east of 39° W, the conductivity offset west of 39° W is about -0.022 S m$^{-1}$, while east of that is about 0.094 S m$^{-1}$. These differences in conductivity correspond to a change in salinity of -0.015 and 0.08 g kg$^{-1}$, respectively. For probe #70200068, the conductivity offset is indistinguishable from 0 west of 36° W, while east of that is 0.156 S m$^{-1}$, which corresponds to a salinity difference of 0 and 0.15 g kg$^{-1}$. No trend in the offset east of 23.5° W is detected. For the 14.5° N section, we had uCTD, CTD, and TSG data available and the respective calibrations uCTD/CTD and uCTD/TSG could be compared. This was done in order to see if in case only TSG data is available (as it is the case for the 11°S section), still reasonable calibration results could be achieved. For both probes, the TSG derived drifts occurred in the same longitude range as they were detected using the CTD data. However, the magnitude of the



offset was generally smaller for the TSG compare to the CTD-based method, especially for probe #70200126 in the longitude range west of 39° W, where even the sign of the offsets was opposite to each other. Such a difference is likely due to the fact that the CTD-based method employs a specific potential temperature range where the salinity variation is small, while the TSG-based method focuses only at near-surface values (6.5 m), where the salinity varies in a broad range. Therefore, we would trust more the CTD-based method, and note that if the TSG-based method returns a small conductivity offset (<0.03 S m$^{-1}$), one might need more caution to apply this offset to calibrate the uCTD. However, one needs also more caution when applying the CTD-based calibration in regions, where the T/S relation of the central water shows a mixture effect of NACW and SACW. At the 11° S section, CTD data was only available at the beginning and end of the section, we could use only the TSG data as the primary source for validation. Fortunately no drift was detected in the uCTD probes conductivity cell, but a stable offset with a mean value of 0.131 S m$^{-1}$ and 0.073 S m$^{-1}$ was detected and applied for the probes #70200068 and #70200138, respectively.

After the offset/drift calibration, all the uCTD data was gridded vertically from the original resolution (~0.25 m at a nominal sinking speed of 4 m/s) to 1 m for the geostrophic velocity calculation later. We estimate that the calibrated and gridded uCTD data have an accuracy of 0.02-0.05 g kg$^{-1}$ in salinity, and 0.004 °C in temperature (Rudnick and Klinke, 2007).

All calculations in this study are based on the Thermodynamic Equation of State for seawater 2010 (TEOS-10, McDougall and Barker, 2011). TEOS-10 is introduced to replace the previous Equation of State, EOS-80, and it provides a thermodynamically consistent definition of the equation of state in terms of Gibbs function for seawater. The most obvious change in TEOS-10 is the adoption of conservative temperature ($\Theta$) and Absolute Salinity ($S_A$) to replace the potential temperature and practical salinity. Although the new equation of state has a non-negligible effect on the density field in the deep ocean, its effect in the upper ocean is expected to be small; therefore, our results obtained using TEOS-10 should be comparable with the previous studies.



## 2.2 ADCP measurements

Direct current velocity profiles were measured continuously during all three cruise legs with vessel-mounted 75 kHz and 38 kHz Teledyne RDI Ocean Surveyors (OS75 and OS38). The OS75 was configured to measure at a rate of 2.2 s and a bin size of 8 m. The measurement range varied between

500 m and 700 m. The OS38 was set to measure at a rate of 3.5 s and at 16-m (32-m) bin size during 14.5° N (11° S) section. The measurement range was mostly 1200 m. Ship navigation information was synchronized to the ADCP system. The misalignment angles and amplitude factors were calibrated during post-processing. The processed data contain 10-minutes averaged absolute velocities in earth coordinates; the first valid bin for OS75 is 18 m at 14.5° N and 13 m at 11° S, for OS38 is 21 m at both

sections. In this study, only the OS75 velocity was used since it has a higher accuracy in upper layers and higher vertical resolution. The uncertainties of 1-h averages were estimated by Fischer et al. (2003) to be 1–3 cm s$^{-1}$.

## 2.3 Wind data

We used three different wind datasets in our analysis. First, we used the observed wind speed and

direction recorded with the R/V Meteor anemometer, mounted at a height of 35.3 m. The wind data were stored with a temporal resolution of one minute. True wind speed and direction were calculated using ship speed and direction from navigation system. On-station measurements were removed. The reduction from observation height to 10 m standard height was calculated according to Smith (1988) and wind stress was calculated according to Large and Yeager (2004) assuming neutral stability. The

final wind stress used for the Ekman transport calculation was binned in 50-km ensembles to filter out small-scale variability.

The blended Satellite-based level-4 Near-Real-Time wind stress product (hereafter satellite wind stress) from Copernicus Marine Environment Monitoring Service (CMEMS) was used. The wind speed data is derived from retrievals of scatterometers aboard satellite METOP-A (ASCAT) and Oceansat-2

(OSCAT) and combined with the European Centre for Medium-Range Weather Forecasts (ECMWF) operational wind analysis and gridded to 0.25° × 0.25° resolution in space and 6 hours in time. The wind stress data was estimated using COARE 3 model (Fairall et al., 2003).





Moreover, the NCEP/NCAR monthly zonal wind stress at 14.5° N and 11° S corresponding to the months of the cruises (i.e. May and July 2013) was used to calculated the Ekman transport.

**2.4 GECCO2 ocean synthesis data**

In order to integrate our local observational results into a large scale circulation the GECCO2 ocean

synthesis data was used and compared (Köhl, 2015). GECCO2 is a German version of the MIT general circulation model "Estimating the Circulation and Climate of the Ocean system" (ECCO, Wunsch and Heimbach, 2006). It has $1° \times \frac{1}{3}°$ resolution and 50 vertical levels. GECCO2, includes the Arctic Ocean with roughly 40 km resolution and a dynamic/thermodynamic sea ice model of Zhang and Rothrock, (2000). The synthesis uses the adjoint method to bring the model into consistency with available

hydrographic and satellite data (Köhl, 2015). The prior estimate of the atmospheric state is included by adjusting the control vector, which consists of the initial conditions for the temperature and salinity, surface air temperature, humidity, precipitation and the 10-m wind speeds from the NCEP RA1 reanalysis 1948-2011 (Köhl, 2015). The surface fluxes are derived by the model via bulk formulae of Large and Yeager (2004). For the study period from May to July 2013 monthly and daily output data

was available. It is important to note that the in-situ observational data measured during the cruises were not assimilated in the synthesis, while the satellite measured wind speed was assimilated but possibly modified via the synthesis.

**3 Results and discussion**

**3.1 Upper layer hydrography at 14.5° N and 11° S**

Along both sections (Fig. 2a, b) the typical upward tilting of isotherm towards the east, as a result of the subtropical gyre circulation, can be seen. Along nominal 14.5° N the water in the upper 50 m, compared to that at 11° S, was relatively warm and fresh, with an averaged $\Theta$ and $S_A$ of about 26.03 °C and 36.15 g kg$^{-1}$, respectively. The minimum $S_A$ core near the western boundary probably originates from the freshwater runoff from the Amazon River (Fig. 2c). Together with the warm temperature, it forms the

lightest water observed along the section (Fig. 2e). A subsurface salinity maximum layer of Subtropical





Underwater (STUW) is centred at 100 m depth with $S_A$ greater than 37.2 g kg$^{-1}$. STUW is formed in the Subtropical Atlantic with Sea Surface Salinity (SSS) maximum due to excessive evaporation, and subducted equatorward (Talley et al., 2011). The upward tilt of the isopycnals from west to east is suggestive of a net southward geostrophic transport when excluding the western boundary, where sharp

deepening of the isopycnals implies a northward, intensified boundary current (Fig. 2g).

Because the ocean is not homogenous, a control surface must be defined that characterizes the maximum penetration depth of the momentum flux into the upper ocean. One choice would be the MLD, which we defined as the depth where the density increased by 0.01 kg m$^{-3}$ in reference to the value at 10 m (following Wijffels et al. 1994). Along the 14.5° N section, this MLD is relatively

shallow (on average 25.1 m), and as such unlikely a representative of $D_E$. According to the Ekman theory, $D_E$ for water at 14.5° N with a typical vertical eddy viscosity, $A_v$ of 0.02 m$^2$ s$^{-1}$ would be 33.1 m (see the definition of $D_E$ in Eq. 1).

Alternatively a top of the pycnocline (TTP) has been defined as the shallowest depth at which the density gradient is larger than 0.01 kg m$^{-4}$ (Wijffels et al., 1994). The TTP is typically deeper than the

MLD and better defines the transition depth between well-mixed and stratified ocean, up to which the momentum from the wind is transferred (Chereskin et al., 2002). At some locations along both sections we observed two homogenous layers of slightly different density and possibly a remnant of the seasonal mixed layer cycle. In these cases, the TTP depth was chosen as the deeper one of the depth that satisfies the density gradient criterion. On average along the 14.5° N section, this TTP depth is 45.8 m. At both

ends of the section, the TTP coincided with the MLD and was relatively shallow, while in the rest part of the 14.5° N section, TTP was deeper than the MLD. Since TTP was defined based on a gradient criterion, it represents the bottom of a weakly stratified surface layer rather than a specific density surface.

At 11° S, the surface water was cooler and more saline than that at 14.5° N with an averaged $\Theta$ and $S_A$

of about 24.52 °C and 36.69 g kg$^{-1}$. The STUW with maximum salinity larger than 37.3 g kg$^{-1}$ was centred at about 100 m but even saltier than that at 14.5° N. Likewise, a net northward geostrophic flow can be anticipated from the displacement of the isopycnals. At the western boundary, the North Brazil



Undercurrent (NBUC) is characterized by a narrow and strong northward velocity band west of 35° W (Fig. 2h). Along 11° S, the mean MLD and mean TTP depth is 32.2 m and 56.8 m, respectively, generally deeper than that at 14.5° N.

In the hydrographic data $\Theta/S_A$ variability is seen at both sections that are associated with mesoscale
eddies. For instance, at 14.5° N/25° W, and 11° S/7° E, were cyclonic and anticyclonic eddies characterized by the upward peak of the isotherms, and were clearly visible from the geostrophic velocity sections (Fig. 2g, h).

The daily $\Theta$ and $S_A$ data of the GECCO2 synthesis were extracted from the model grid to the nearest time and position of the ship measurement. In general, GECCO2 daily data reproduced the observed
hydrographic structure very well (not shown). The upward tilt of the isopycnals from the west to the east and the subsurface salinity maximum with $S_A$ larger than 37.2 g kg$^{-1}$ were clearly captured by GECCO2. However, the most obvious difference was at the western boundary of 11° S, where the surface salinity was not as high as the observed values, and the isopycnals were not tilting in the same direction, indicating that the shallow western boundary current in the GECCO2 flew in the opposite
direction compared to the observation at 11° S. But we expect that this difference should not impact the ageostrophic velocity calculation, since the geostrophic velocity must be removed from the total velocity.

### 3.2 Vertical structure of the ageostrophic flow
At 14.5° N both CTD and uCTD measurement were done throughout the section and a geostrophic
velocity was calculated from these two datasets independently. For CTDs, the relative geostrophic velocity referenced to 200 m was computed between the adjacent stations (average distance about 75 km). For uCTDs, in order to take the advantage of the high spatial resolution, the relative geostrophic velocity to 200 m was calculated between any closest pair of uCTD profiles with a minimum distance of 70 km (roughly the Rossby radius of deformation at this latitude). To obtain the absolute geostrophic
velocity, the reference velocity at 200 m was obtained from the ADCP measurement. The ADCP velocity data was projected to the normal direction of the cruise track and then averaged between the corresponding CTD/uCTD pairs. We did not include the ADCP velocity data recorded at the CTD



stations, because velocity was repeatedly measured at a CTD station, zonally averaging the ADCP velocity would bias the result towards the on-station velocity. At 11° S section, CTD profiles were only taken in the vicinity of the coasts, and over most of the section only uCTD data is available (Fig. 1). Therefore the geostrophic velocity was computed from the combined CTD and uCTD dataset following

the methodology applied to uCTD data at the 14.5° N section, except that at 11° S the minimum distance between the closest profiles was set to 90 km (roughly the Rossby radius of deformation at 11° S). Note that the distance of uCTD profiles for geostrophic velocity calculation is an arbitrary choice, varying the distance from 70 km to 110 km makes a negligible effect on the total transport (less than 2%).

The ageostrophic velocity was then calculated as the difference between the ADCP velocity and absolute geostrophic velocity. In previous studies (Wijffels et al., 1994; Chereskin et al., 1997; Garzoli and Molinari, 2001), the corresponding ADCP velocity at the reference depth was taken as the reference velocity, assuming that the flow at the reference depth was in geostrophic balance. However, the section-averaged ADCP velocity profile for the 14.5° N section show a complicated vertical structure

(Fig. 3a) and it is not obvious at which depth the flow is approximately in geostrophic balance. Thus, referencing the relative geostrophic velocity to the ADCP velocity only at a chosen depth may lead to a biased absolute geostrophic velocity. As a result, the ageostrophic velocity may be sensitive to the choice of the reference level. To overcome this problem, a reference velocity was calculated as an averaged offset between each relative geostrophic velocity and the corresponding ADCP velocity within

a common depth range, over which the ageostrophic components are averaged to about 0. This averaged offset should represent the absolute geostrophic velocity at the reference depth and is roughly independent of the vertical variation due to the ageostrophic components. At 14.5° N, the common depth range for the CTD-based calculation is between 70 m to 500 m, which is below the surface Ekman layer and covered by both CTD and ADCP measurement, while for the uCTD-based calculation,

the depth range is between 70 m and 250 m, since the uCTD was only deployed to measure the upper 250 m depth. At 11° S the depth range is between 100 m and 300 m, which should be also below the Ekman layer and covered by the uCTD and ADCP measurement. Note the choice of the depth range still affects the reference velocity due to the vertical variation in the ADCP meridional velocity, for



example, using a depth range between 70 and 250 m for the CTD based calculation (same as the uCTD depth range) would decrease the final ageostrophic velocity by 0.44 cm s$^{-1}$, using other depth range would not result in an absolute difference exceeding this value. This is much smaller compared to uncertainty caused by using the ADCP velocity at a single depth as the reference velocity (up to 1.75

cm s$^{-1}$), as can be anticipated from the section averaged meridional ADCP velocity (Fig. 3a). The sensitivity of the absolute geostrophic velocity to the choice of the reference level was also tested at 14.5° N. Changing the reference level from 150 m to 250 m would make a change in the absolute geostrophic velocity indistinguishable from zero.

Although northward (southward) ageostrophic velocity at 14.5° N (11° S) dominates the upper 50-70 m

(Fig. 2i, j), as expected from the persistent westward trade winds, the appearance of southward (northward) velocity at 14.5° N (11° S) in the upper 50-70 m and below indicates the existence of non-Ekman ageostrophic component. This will be discussed in details below. The section-averaged ageostrophic velocity based on CTD data at 14.5° N shows a relatively complicated vertical structure with multiple maxima and minima (Fig. 3a). It has a northward maximum velocity of 3.5 cm s$^{-1}$ near the

surface, and decreases to about 0.3 cm s$^{-1}$ at about 60 m, followed by a minor peak at about 80 m before approaching 0 at 100 m. Another peak of 1 cm s$^{-1}$ appears at about 150 m, and below 180 m the velocity changes direction. When the geostrophic velocity is calculated based on uCTD data instead of CTD data, due to the much higher horizontal resolution, the geostrophic velocity field is noisier than that based on CTD data, but the basin-scale structure of the circulation is very similar. For instance, at the

western boundary energetic northward cross-section flow is captured by both CTD and uCTD-based geostrophic velocity. Besides, in order to obtain the absolute geostrophic field, the relative geostrophic velocity from both datasets must be constrained by a reference velocity that is from the same shipboard ADCP measurement, which guarantees the consistency between the geostrophic velocity based on both datasets. As a result, the uCTD-based ageostrophic velocity has a very consistent structure and strength

compared to the CTD-based ageostrophic velocity (Fig. 3a). This is meaningful information as the hydrographic data at 11° S consists primarily of uCTD data. The good agreement between the CTD and uCTD data analysis at 14.5° N justify the use of either one or the other. At 11° S, the ageostrophic velocity shows a near-surface southward maximum of 4.3 cm s$^{-1}$ and decreases almost linearly in the



upper 70 m and gradually approaches 0 at about 100 m (Fig. 3b). In contrast to the northern section the vertical variations of the ageostrophic velocity profile below 100 m are very small.

Assuming that the Ekman balance would hold true along the analyzed sections, the ageostrophic velocity would decrease undisturbed from its surface maximum to about 0 at a certain depth (Ekman depth, $D_E$). However, the observed wave-like ageostrophic vertical structure at 14.5° N indicates that other processes must play a role in setting the section mean ageostrophic flow field. To identify where this wave-like structure came from we first investigated the ADCP velocity. Since the ADCP measurement should capture all the velocity components, the priority was to separate the non-Ekman ageostrophic flow from the other components. A residual velocity was calculated by subtracting an 80-m boxcar filtered velocity profile from the original ADCP meridional velocity. The 80-m filter window was determined based on the vertical length scale of the wave-like structure in the section-averaged ageostrophic velocity profile by visual inspection. At 14.5° N, vertically alternating structures with wavelength of 60 to 80 m are clearly visible, they are coherent and persistent throughout the section, and are most pronounced between 52° W and 46° W (Fig. 4a). At 11° S, similar signals are visible for most of the section, but not as strong as at 14.5° N (Fig. 4b).

Zonally averaging the residual velocity results in a velocity profile with vertically alternating structure similar to that in the section-averaged ageostrophic velocity in both strength and structure, indicating that the vertical variation in the ageostrophic velocity mainly arises from the presence of high order baroclinic waves. Figure 4c and 4d show the buoyancy frequency ($N^2$) calculated from a combination of CTD and uCTD data for the two sections, respectively. It appears that the wave-like signals occur mainly in strongly stratified layer (pycnocline) marked by high $N^2$ values.

Chereskin and Roemmich (1991) also observed energetic, circularly polarized, relative current of large horizontal coherence below the base of the mixed layer at 11° N in the Atlantic. Correspondingly, a secondary maximum appeared at 150 m in their section-integrated ageostrophic transport (analogue to the section-averaged ageostrophic velocity). They described the signal as the propagation of near-inertial internal waves, and argued that the presence of near-inertial peak in internal wave spectra, together with continuous variable wind forcing would guarantee the appearance of these waves. Using





satellite based wind stress data, we tried to examine the changes of wind stress at the measurement points within the last two weeks before the ship arrived at the measurement point. Although, the wind stress shows strong changes in the last two weeks before the ship's arrival for the whole section, it is still not indicative why the wave signal is strongest between 52° W and 46° W. It is tempting to believe

that these waves are near-inertial internal waves, as they mainly exist below the mixed layer, are coherent at large horizontal length scales, and occurred after varying wind field. However, due to the fact that the ship moved nearly constantly except when it was on station, it is extremely difficult to identify what exactly these signals are.

An attempt was made to remove these wave-like signals from the total ageostrophic velocity, since the

non-Ekman ageostrophic components, $v_{ageo\,nonEkman}$, are the primary source of error in the direct Ekman transport estimation (next section). A residual velocity was calculated similar to that in Fig. 4a and 4b, except that a boxcar filter was only applied below the local TTP, since the wave structure was mainly present below the TTP. This residual velocity was then averaged between the CTD/uCTD pairs that were responsible for geostrophic velocity calculation, and then subtracted from the corresponding

ageostrophic velocity profile. At 14.5° N, the section-averaged wave-removed ageostrophic velocity profile shows indeed much less vertical wave structure in the upper 150 m, but also its strength in the upper 50 m slightly decreased (Fig. 3a). Interestingly, below 200 m, there are still ageostrophic flows of 1 cm s$^{-1}$ magnitude with vertical length scale of several hundred meters. This signal has not been identified, but it is much deeper than the expected Ekman depth, hence it is not considered in this work.

At 11° S, the minor variation below 100 m is nearly completely removed by this approach, the ageostrophic velocity decrease smoothly to zero at depth (Fig. 3b). But, the ageostrophic velocity in the upper 70 m appears being disturbed by the removal of the wave signal. This impact is expected, since the structure of the wave-removed ageostrophic velocity could also be affected by the large variation of TTP depth between the eastern and western basin, when it was zonally averaged. The boxcar filter

presented here is only an example of separating the baroclinic structure from the total velocity. More sophisticated methods may be applied, for instance, Smyth et al. (2015) took the Doppler shift in the shipboard current measurement into account, and translated observed Yanai wave properties into the reference frame of the mean zonal flow. But this is obviously beyond the scope of this work.



### 3.3 Ekman transport

### 3.3.1 Indirect method

According to Eq. 1, the Ekman transport can be calculated from the wind stress data (referred as indirect method) by integrating the left-hand side of Eq. 1 zonally. The in-situ wind stress data and a satellite-

based wind stress product from CMEMS were used. The satellite wind stress data were extracted from the original grid to the nearest time and nearest position of the ship navigation. Both in-situ and satellite wind stress were projected to the tangential direction of the cruise track, so that the cross-section Ekman transport at each grid point was calculated. Note that both sections were occupied nominally zonally, therefore, we will refer cross-section Ekman transport as meridional Ekman transport for simplicity

hereafter.

Overall, the satellite wind stress agrees well with the ship wind stress (Fig. 5) except of the region between 40° W and 30° W at 14.5° N, where the zonal ship wind stress is larger than the zonal satellite wind stress and at 11° S the ship wind stress is generally smaller than the satellite wind stress. Since the 10-m wind speed between the ship and satellite is very close to each other at both sections (not shown),

the difference in the wind stress is likely due to use of different drag coefficient formulation (COARE 3 for CMEMS wind product; Large and Yeager, 2004 for ship wind stress). This effect will become more obvious, when the meridional Ekman transport is estimated using the satellite and ship wind stress. In comparison to the NCAR/NCEP monthly zonal wind stress, the weaker ship wind stress at the western half of the 14.5° N section indicates that the cruise started with anomalous weak winds, while at 11° S

the observed wind stress (both ship and satellite observation) were consistent with the monthly mean wind stress.

As expected, at 14.5° N, the indirect estimate of the Ekman transport from the in-situ wind stress is 6.7 ± 3.5 Sv, only 0.4 Sv larger than that from the satellite wind stress. Using the monthly mean wind stress from NCEP/NCAR during the M96/M97 cruise month (May 2013), the total transport is 8.8 Sv. The

difference between the monthly wind estimate and in-situ wind estimate is mainly due to the anomalously weak wind when the cruise started from the western boundary (Fig. 5a). At 11° S, the indirect Ekman transport from the in-situ wind stress is 13.6 ± 3.3 Sv, while the transport from the



satellite wind stress is 2.0 Sv higher, due to the higher value of the satellite wind stress (Fig. 5b). The NCEP/NCAR monthly wind stress in July 2013 returns a transport of 15.3 Sv. The errors shown with the indirect ship wind estimates are given by the standard deviation of the long-term Ekman transport calculated using 6-h NCEP/CFSR wind stress between the years 2000 and 2011 at the two latitudes.

Another source of uncertainty may arises from the wind stress calculation using different bulk formulas, which could lead to an uncertainty as large as 20% (Large and Pond, 1981). This may explain the difference in the indirect estimates between using in-situ wind stress and the satellite wind stress at 11° S.

### 3.3.2 Direct method

The direct meridional Ekman transport is derived from vertically integrating the ageostrophic velocity profiles (Eq. 1, right-hand side). As already mentioned, one critical assumption is the integration depth ($D_E$). Applying the TTP as an estimate of $D_E$, the total Ekman transport at 14.5° N based on CTD data is 6.2 ± 2.3 Sv, while applying a uniform depth of 50 m results in an alternative estimate of 6.5 ± 1.9 Sv, and applying the local MLD results in a transport of 5.1 ± 1.4 Sv. When integrating the ageostrophic

velocity calculated from the uCTD data to the TTP, the Ekman transport is 6.6 ± 2.3 Sv. At 11° S, the direct estimate by applying the TTP, a uniform depth of 80 m, and the MLD is -11.7 ± 2.1 Sv, -12.0 ± 2.4 Sv, and -8.0 ± 1.4 Sv, respectively ("-"denotes southward transport). Note that the Ekman transport estimates listed above were calculated using the original ageostrophic velocity. Integrating the wave-removed ageostrophic velocity at 14.5° N to the TTP depth would result in an estimate indistinguishable

from the estimates using original ageostrophic transport, since the boxcar filter was only applied below the local TTP depth. The errors given with the transport estimates were calculated following Chereskin and Roemmich, (1991) and Wijffels et al., (1994). Assuming that near-inertial motions are the dominant source of error, decorrelation length scales were calculated as the distance that the ship travelled in a quarter of the inertial period at 14.5° N (47.9 hours) and 11° S (62.7 hours) resulting in 130 km and 230

km, respectively. In total, 38 segments of the 14.5° N section and 25 segments of the 11° S section were obtained by dividing the total distance of each section by the corresponding decorrelation length scale, respectively. The westernmost and easternmost 4 segments of each section were omitted, because the anomalously weak wind near the eastern boundary, and the strong boundary current in the western





boundary region. The degree of freedom (DOF) of 30 and 17, respectively, was the number of the remaining segments. The ageostrophic transport within each segment was treated as an independent realization of the Ekman transport. Therefore, standard errors were calculated. Then the final error is given as the standard error times the DOF. Another factor that could lead to an uncertainty is the depth

range used to calculate the reference velocity from the ADCP velocity. As discussed above, we argue that the vertical structure of the ageostrophic velocity should arise from the near-inertial motion, therefore, should be included already in this uncertainty estimate.

The shallowest valid bin depth of the ADCP measurement was 18 m (13 m) at 14.5° N (11° S), the ageostrophic velocity was extrapolated linearly to the surface using the value of the first two bins. Note

that we did not assume a surface maximum of the ageostrophic velocity everywhere, since for individual profiles the ageostrophic velocity at the first bin depth may be smaller than that at the second bin, which would result in a smaller surface ageostrophic velocity. In previous studies (Chereskin and Roemmich, 1991; Wijffels et al., 1994), the velocity above the first ADCP bin was assumed constant as the value at the first bin. Using this assumption would reduce Ekman transport at 14.5° N by 0.56 Sv

(9% of total northward transport), and at 11° S by 0.14 Sv (1% of total southward transport). According to the classical Ekman theory, the surface Ekman velocity ($V_0$) is 45° to the right (left) of the wind blowing direction in the northern (southern) hemisphere and can be derived from the total wind stress (see the definition of $V_0$ in Eq. 1). As a comparison to the linear extrapolation above the first ADCP bin, we also calculated the meridional Ekman velocity at the surface using the total in-situ wind stress and a

constant $A_v$ of 0.02 m²s⁻¹. Then the meridional ageostrophic velocity above the first ADCP bin was linearly interpolated using the value at the first bin and the surface meridional Ekman velocity predicted from the in-situ wind stress. The resulting Ekman transport is 1.2 Sv (0.7 Sv) smaller than that using a linear extrapolation method at 14.5° N (11° S). Note that we chose a linear extrapolation method, because it resulted in a better agreement between the indirect and direct estimates, but it may

overestimate the total ageostrophic transport.

A question followed is whether the ageostrophic flow in the mixed layer has shear or is constant with depth referred to a slab-like shape. Given the large variation of the MLD throughout the sections, basin-wide averages are not conclusive. Chereskin and Roemmich (1991) found shear structure in the mixed





layer at 11° N in the Atlantic, while Wijffels et al. (1994) reported a slab-like shape at 10° N in the Pacific, and attributed the shear structure found by Chereskin and Roemmich (1991) to an improper definition of MLD. Following their method, the depth was normalized by the local MLD before averaging the ageostrophic velocity across the basin (Fig. 6). At 14.5° N, for a slab-like ageostrophic

structure, Fig. 6 would show a nearly constant profile from the surface to about the MLD. Instead, it shows strong shear above the MLD. Such strong shear is insensitive to the definition of MLD. For example, choosing a density threshold of 0.005 kg m$^{-3}$, the shear still exists below 0.4 MLD. At 11° S, no slab-like structure in the ageostrophic velocity was found, either. The constant value above 0.3 MLD is a consequence of using a constant velocity above 18 m, the shallowest ADCP bin. Therefore, we

would conclude that ageostrophic shear exists within the mixed layer in our cases, as expected from the classical Ekman theory.

The cumulative Ekman transport from the western to the eastern boundary shows an overall match between the direct and indirect methods (Fig. 7a, b). At 14.5° N, the in-situ wind was relatively weak at the beginning and the end of the section. Correspondingly, the increment in transport within these two

segments was moderate, while in the central part of the section, where the wind was strong, the rapid accumulation of Ekman transport is directly visible in both indirect and direct estimates. The direct estimates using TTP and 50-m depth are very close to the in-situ wind estimates. The estimate using 50-m depth tends to overestimate the transport close to both ends of the section. Applying the MLD as integration depth tends to underestimate the total transport by about 1.5 Sv, compared to the ship wind

estimate. This is mainly because it fails to capture the increase between 30° W and 25° W. Note that the uCTD-based direct estimate is consistent with the CTD based estimates, though it overestimates the transport in the middle of the section, the total transport as well as the transport structure is similar. This may be a result of the higher spatial resolution of the uCTD measurement, which captures more details in the horizontal features introduced by the wind.

At 11° S, the wind was strong in the first half of the basin, then gradually weakened when approaching the eastern boundary. Correspondingly, the Ekman transport accumulates rapidly to about 12 Sv at 0° E, east of which the increment is very small for both direct and indirect estimates. Among the direct estimates, integrating the ageostrophic velocity to 80 m and TTP returns nearly identical transport in the





western half of the section; the difference in the eastern half mainly reflects the shallower TTP towards the eastern boundary, while using the MLD for the integration underestimates the Ekman transport from the very beginning. Note that at both sections, the direct estimate using MLD is about one-fourth smaller than that using TTP depth. This agrees with the findings at 10° N in the Pacific by Wijffels et al.

(1994), who reported that the Ekman flow within the mixed layer accounted for about two-third of the total Ekman transport, and the in-situ wind predicted the Ekman transport down to the TTP.

### 3.4 Ekman transport from GECCO2

The daily data of the GECCO2 synthesis (2008 to 2014) allowed us to estimate the model Ekman transport, inspect the vertical structure of the ageostrophic velocity in the model and compare these

results with the observation for the corresponding cruise time periods. The daily data were first extracted from the model grid to the nearest ship time and position. The Ekman transport in GECCO2 was calculated in a similar manor as the direct method used for the observational data. An ageostrophic velocity was calculated as the difference between the geostrophic velocity and total velocity with a reference depth of 200 m. The geostrophic velocity was computed from the temperature and salinity of

GECCO2. The direct estimate of the meridional Ekman transport in GECCO2 was obtained by integrating the ageostrophic velocity vertically and zonally.

The section-averaged ageostrophic velocity at both sections shows a near surface maximum at about 15 m, then decreases sharply to 0 at about 50 m, the flow is purely geostrophic below 60 m (not shown). This vertical distribution of ageostrophic velocity indicates that wind-driven Ekman component is the

predominant contributor to the ageostrophic velocity in the GECCO2 model, and that nearly all the wind energy is utilized for the Ekman transport and confined to the upper 50 m at both sections. The total transport by integrating the ageostrophic velocity to 50 m is 7.6 Sv at 14.5° N, and 12.0 Sv at 11° S, respectively (Fig. 7), which is close to the indirect Ekman transport estimate based GECCO2 daily wind stress of 7.4 Sv and 13.4 Sv, respectively.

This result agrees very well with the observed direct Ekman transport, which is likely due to the fact that GECCO2 daily wind stress has similar magnitude to the in-situ wind stress. The observed ageostrophic cumulative transport shows strong mesoscale fluctuation throughout the sections, this is



characterized by the presence of northward and southward ageostrophic velocity even though the in-situ wind is persistently westward, while the GECCO2 ageostrophic transport accumulates smoothly (Fig. 7).

### 3.5 Ekman heat and salt flux

The Ekman heat and salt fluxes, $H_e$ and $S_e$ respectively, were calculated using the direct method (ageostrophic velocity approach) and the indirect method (wind stress approach). In the previous sections, we have discussed that the TTP was a reasonable assumption for the depth of $D_E$ for both sections. Hence, using the $\Theta$ and $S_A$ together with the ageostrophic velocity within the layer from the sea surface to the TTP (referred to as TTP layer) should give us the best estimate of the heat and salt

fluxes (referred as: direct TTP/profile). As a comparison, the fluxes using only in-situ SST from the CTD and uCTD were also calculated (referred as direct TTP/surface).

Often Ekman heat and salt fluxes are estimated by combining the Ekman volume transport inferred from wind stress with the SST and SSS from a climatology or satellite measurements (e.g. McCarthy et al., 2015). Here, we calculated the heat flux using in-situ wind and in-situ SST data (referred as indirect

surface) to compare with the direct estimates and previous studies. Wijffels et al. (1996) assumed a linear Ekman velocity profile between the surface and TTP and calculated the Ekman heat and salt fluxes using climatological wind stress data, combined with the in-situ temperature and salinity. Here we follow their method and use the in-situ $\Theta$, $S_A$ and wind to calculate the Ekman heat and salt fluxes (referred as indirect TTP) as a counterpart to the direct TTP method. Additionally, an annual Ekman

heat and salt flux (referred as indirect annual) were calculated using an annual average of the monthly NCEP/NCAR reanalysis wind stress data between 1991 and 2013 and the annual average of SST and SSS from the Roemich-Gilson monthly Argo climatology (Roemmich and Gilson, 2009, hereafter RG climatology).

Note that in order to calculate the Ekman flux in the context of mass conservation (Montgomery, 1974),

it has to be assumed that the Ekman volume transport in the upper layer is balanced by an equal and opposite geostrophic return flow at depth. This is a reasonable assumption and has been routinely adopted in many inverse studies (Ganachaud and Wunsch, 2003). To account for this return flow, an





averaged conservative temperature, $\overline{\Theta}$, and absolute salinity $\overline{S_A}$ was subtracted from the in-situ $\Theta$ and $S_A$ at each section. $\overline{\Theta}$ and $\overline{S_A}$ are the zonal and vertical (0-5000 m) average of the conservative temperature and absolute salinity, calculated from the annual climatology of World Ocean Atlas 2013 v2 (Locarnini et al., 2013; Zweng et al., 2013) at each section.

It is worth to note that the Ekman salt flux presented in this study may not be very representative by its own but it may provide a quantitative measure of the sensibility of Ekman salt flux to the change in Ekman volume transport and Ekman salinity. This would become significant when setting the constraints for salt conservation and the resulting freshwater flux in the further studies of meridional overturning circulation in the same region. The details of each method are described below. Note that

only Ekman heat flux and transport weighted Ekman temperature are shown here. The salt flux and transport weighted Ekman salinity are analogue.

### 3.5.1 Direct methods

The Ekman heat flux $H_e$ were calculated using the direct TTP/profile method as

$$H_e = \rho C_p \int_{x_1}^{x_2} \int_{-TTP}^{0} (\Theta(x,z) - \overline{\Theta}) v_{ageo}(x,z) dz dx, \tag{3}$$

where $C_p$ is the specific heat capacity of sea water at constant pressure, $\rho$ is the density of sea water, in this study we assumed a constant $C_p = 4000$ J kg$^{-1}$ °C$^{-1}$ and a constant $\rho = 1025$ kg m$^{-3}$, $v_{ageo}$ is the ageostrophic velocity, $\Theta$ is the in-situ conservative temperature. $\overline{\Theta}$ is the mean conservative temperature at the corresponding section.

It is useful to consider the Ekman flux as the product of the Ekman mass transport and the transport-

weighted temperature. The Ekman temperature then can be calculated as follows.

$$\Theta_E = \frac{\int_{x_1}^{x_2} \int_{-TTP}^{0} \Theta(x,z) v_{ageo}(x,z) dz dx}{M_{direct}^{y}}, \tag{4}$$

The direct surface calculation (direct TTP/surface) is similar to the direct TTP/profile method, for which, $\Theta(x,z)$ in Eq. 3 and Eq. 4 is replaced by its surface value $\Theta(x, z = 0)$.



Following Chereskin et al. (2002), to estimated the uncertainties of the direct Ekman heat and salt flux, only northward or southward ageostrophic velocity were used in both numerator and denominator in Eq. 3 and Eq. 4 for 14.5° N or 11° S section, respectively. Since the wind direction was predominantly uniform and westward, the uncertainty should mainly caused by the ageostrophic velocity that was

opposite to the expected Ekman flow direction.

### 3.5.2 Indirect surface method

Following Levitus (1987), the Ekman heat flux for the indirect surface method was calculated as

$$H_e = C_p \int (\Theta(x, z = 0) - \overline{\Theta}) \frac{\tau_x}{f} dx, \tag{5}$$

where $\tau_x$ is the in-situ wind stress in the tangential direction of the cruise track, $f$ is the Coriolis

parameter. The transport-weighted temperature is calculated as follows,

$$\Theta_E = \frac{\int \Theta(x, z = 0) \frac{\tau_x}{\rho f} dx}{M^y_{indirect}}, \tag{6}$$

The indirect annual method is an analogue to the indirect surface method, except, the Ekman volume transport and SST were derived from the NCEP/NCAR reanalysis wind stress and RG climatology, respectively.

### 3.5.3 Indirect TTP method

Following Wijffels et al. (1996), the Ekman heat flux for the indirect TTP method was calculated with the assumption of a linear weighting of the temperature and salinity with depth.

$$H_E = C_p \int [\frac{2}{3}\Theta(x, z = 0) + \frac{1}{3}\Theta(x, z = TTP) - \overline{\Theta}] \frac{\tau_x}{f} dx, \tag{7}$$

where $\Theta(x, z = TTP)$ is the in-situ conservative temperature at TTP depth from the CTD/uCTD. The

transport-weighted temperature is calculated as follows,

$$\Theta_E = \frac{\int [\frac{2}{3}\Theta(x, z = 0) + \frac{1}{3}\Theta(x, z = TTP)] \frac{\tau_x}{\rho f} dx}{M^y_{indirect}}, \tag{8}$$





### 3.5.4 Results

The Ekman volume, heat and salt fluxes calculated using different methods are summarized in Table 1. It is clear that the differences in Ekman volume transports dominate the differences in resulting Ekman heat and salt fluxes. The higher Ekman volume transport by the indirect methods leads to higher heat

and salt fluxes compared to the direct methods at both sections. At 14.5° N, the transport-weighted Ekman temperature from the direct surface method is 0.10 °C higher than that from the direct TTP method. This temperature difference corresponds to a change in the heat flux by only less than 1 %, which is very small compared to the uncertainty caused by the volume transport uncertainty. The indirect TTC method returns the Ekman temperature and salinity value very close to that of the direct

TTP method, indicating that the assumption of a linear Ekman velocity profile between the surface and the TTP depth may be reasonable. This could be potentially interesting, since this method is independent of the ageostrophic velocity.

At 11° S, the difference between the direct TTP and surface methods is negligible. The transport-weighted Ekman temperature from the indirect TTP and surface methods is significantly smaller than

that from the direct methods. This may be caused by a combined effect of stronger Ekman volume transport by the indirect method near the eastern boundary (Fig. 7), and the cooler water temperature due to coastal upwelling. In other words, the indirect calculation tends to give excessive weighting to the cooler water, which results in lower values in the transport-weighted Ekman temperature. Such a combined effect has limited impact on the Ekman salinity.

The difference in the Ekman heat flux when using temperature at surface or within TTP layer is much smaller than that for the extreme case at the end of the summer monsoon in the Indian Ocean in September 1995 reported by Chereskin et al. (2002). The choice of Ekman temperature and salinity has a negligible effect on the resulting heat and salt flux across the sections studied here.

Note that at 14.5° N (except for the indirect annual method) the Ekman heat fluxes (0.41 to 0.44 PW)

estimated using direct and indirect methods based on ageostrophic velocity and in-situ wind is generally smaller compared with the estimates of 0.7 to 0.8 PW by Levitus (1987) or 0.6 to 0.8 PW by Sato and Polito (2005). As described above, both the direct and indirect methods in this study reflect the Ekman



transport driven by the in-situ wind, which is weak compared to the monthly wind, especially in the western basin (Fig. 5a). By using the annual mean wind stress from NCEP/NCAR reanalysis and SST from RG climatology, the Ekman heat flux is 0.58 PW, which is close to the estimates of Sato and Polito (2005). At 11° S, the direct and indirect Ekman heat fluxes (0.8 to 0.96 PW) are rather close to

the estimate of 1.05 PW by Levitus (1987) or within the range of values (0.7 to 1.0 PW) estimated by Sato and Polito (2005). Here, the Ekman volume transports estimated from in-situ wind and from the annual averaged wind were similar.

### 4 Summary

The Ekman volume, heat and salt transport across zonal sections at 14.5° N and 11° S in the Atlantic

were calculated by using an ageostrophic velocity based method (direct method) and a wind stress based method (indirect method). A cross-section ageostrophic velocity was calculated at each section following Chereskin and Roemmich (1991) and Wijffels et al. (1994) by subtracting the geostrophic velocity from the cross-section component of the ADCP velocity. At both sections, underway-CTD profiles were used for the ageostrophic velocity calculation. A comparison between the results based on

standard CTD und uCTD data at 14.5° N revealed a consistent transport estimate with a robust vertical ageostrophic velocity structure and horizontal distribution of the Ekman transport. This has established our confidence to perform a similar calculation for the 11° S section, along which primarily uCTD profiles were taken.

The section-averaged ageostrophic velocity at 14.5° N and 11° S have a near-surface northward and

southward maximum of 3.5 and 4.3 cm s$^{-1}$, and decreases below to reach about zero at 60 m and 100 m, respectively. This is an indication of the Ekman spiral, and is consistent with the findings of Chereskin and Roemmich (1991) at 11° N in the Atlantic, Wijffels et al (1994) at 10° N in the Pacific and Chereskin et al. (1997) at 8.5° N in the Indian Ocean. Near-inertial oscillations are regarded as a dominant source of ageostrophic noise, which is superimposed upon the wind-driven flow, but zonally

averaging or integration over several inertial periods should remove most of it. However, below the surface-intensified Ekman flow, the ageostrophic velocity along both sections shows wave-like structures of 50-80 m vertical scale with multiple maxima and minima. By applying a boxcar filter,





these wave-like signals were separated from the cross-section ADCP velocity (Fig. 4). The appearance of these structures is mainly below the TTP and coincides with the layer of maximum buoyancy frequency. They are characterized by a horizontal velocity vertically alternating its direction as well as large horizontal coherence. Chereskin and Roemmich, (1991) also reported the presence of such waves

within the main thermocline, which were coherent over large horizontal space scales. These are thought to be near-inertial internal waves. Further studies using moored current observations are expected to deepen our understanding of these wave signals.

The section averaged ageostrophic velocity had its maximum at the depth of the first valid bin of the ADCP (13-18 m), indicating that a shear existed within the ML, despite of its homogeneous density.

Chereskin and Roemmich (1991) examined this at 11° N in the Atlantic by zonally averaging a MLD-normalized ageostrophic velocity, and concluded that shear existed in the ML. However, Wijffels et al. (1994) applied the same technique and found a slab-like layer of ageostrophic velocity above the MLD at 10° N in the Pacific and attributed the discrepancy to a loose definition of the MLD by Chereskin and Roemmich (1991). Following their methods, we also examined whether there was shear in the

ageostrophic velocity within the ML along the two sections. It appears that at both sections, an ageostrophic shear existed in the ML, and this conclusion does not change if a more rigorous constraint on the MLD is used (Fig. 6).

The Ekman transport estimated by integrating the ageostrophic velocity zonally through the section and vertically to the local TTP depth is 6.2 ± 2.3 and 11.7 ± 2.1 Sv at 14.5° N and 11° S, respectively, which

compares reasonably well to the estimates of 6.7 ± 3.5 and 13.6 ± 3.3 Sv by using the in-situ wind stress data. By using a fixed integration depth of 50 m at 14.5° N and 80 m at 11° S, the ageostrophic Ekman transport is not significantly different from those calculated using the TTP depth, while using the MLD as the integration depth, the ageostrophic Ekman transport is about one-forth smaller than using the TTP depth. This is an indication that the wind-driven flow penetrates beyond the ML to the TTP, and it is

consistent with the findings of Wijffels et al. (1994), who reported that two-third of the wind driven transport was within the ML and that the TTP is a reasonable choice for the integration depth of the Ekman flow. Note that above the first ADCP bin (13-18 m), the meridional ageostrophic velocity was linearly extrapolated using the values from the first two bins. However, when the surface meridional





Ekman velocity is assumed equal to the velocity of the first measured ADCP bin (constant extrapolation), or extrapolated toward the theoretical Ekman solution for the surface velocity, the total ageostrophic transport would be up to 1.2 Sv smaller than the results shown above. Therefore, the linear extrapolation may to some extend overestimate the ageostrophic transport.

Between the two sections, the poleward ageostrophic divergence is 17.9 Sv, the equatorward geostrophic convergence in the TTP layer is 6.2 Sv. This result agrees with the conclusion by Schott et al. (2004), who stated that the poleward Ekman divergence induced by the trade winds in both hemispheres is compensated by an equatorward convergence due to the geostrophic flow in the upper layer, but the compensation is generally assumed being not strong enough to reverse the Ekman
divergence.

The cumulative Ekman transport shows rapid increase in the middle of the section and very little changes in the last quarter near the eastern boundary at both latitudes. This is because the meridional trade winds are generally strong and persistent in the western and middle part of the basin, while relatively weak and unstable in strength and direction near the eastern boundary. Similar horizontal
characteristics of Ekman transport were also seen at 11° N and 6° S in the Atlantic (Chereskin and Roemmich, 1991; Garzoli and Molinari, 2001), 10° N in the Pacific (Wijffels et al., 1994). The GECCO2 ocean synthesis daily data were also used to calculate the meridional Ekman transport at 14.5° N and 11° S in the Atlantic by using the ageostrophic approach, which agrees very well with the observed results in respect to horizontal transport structure throughout the basin and the total transport
amount. This was mostly due to the fact that GECCO2 assimilates the observed wind, and with a daily temporal resolution, it is possible for GECCO2 to reproduce the strength of the in-situ wind, thus the Ekman transport. This good agreement has lent us more confidence to use GECCO2 as a reference in the further studies on the MOC at the same latitudes.

An Ekman layer temperature and salinity must be assigned when calculating the Ekman heat and salt
fluxes. Our results suggest that using the SST and SSS for the meridional Ekman heat and salt flux calculation at the two sections is only marginally different from calculations using the temperature and salinity in the TTP layer. It is rather the uncertainty in the Ekman volume transport estimates that



dominates the uncertainties in the Ekman heat and salt fluxes. This is in good agreement with the finding at 10° N in the Pacific by Wijffels et al. (1994), while in striking contrast to that at 11° N in the Atlantic by Chereskin and Roemmich (1991) who found the transport-weighted Ekman temperature is 1 °C cooler than the surface value. The reason for such a contrast is not clear, but it is possible that in

their case the TTP was much deeper than the MLD, especially in the western half of the basin.

Since Ekman volume, heat, and salt transport are significant upper layer components of the MOC in respect to the mass, heat and freshwater conservation, further studies on the vertical and horizontal structure of the Ekman flow, as well as on the Ekman heat and salt fluxes are expected to deepen our understanding and facilitate the studies on the MOC. This study would also provide some reference for

the following up studies on the MOC at the same latitudes.

### 5 Data availability

The Level-4 Near-real-time wind stress product is available at http://marine.copernicus.eu/. The NCEP/NCAR monthly wind stress data is available at http://rda.ucar.edu/datasets/ds090.2/. The NCEP/CFSR 6-h wind stress data is available at http://rda.ucar.edu/datasets/ds093.0/. The Reomich-

Gilson monthly Argo climatology is available at http://sio-argo.ucsd.edu/RG_Climatology.html. The GECCO2 ocean synthesis is available at http://icdc.cen.uni-hamburg.de/1/projekte/easy-init/easy-init-ocean.html#c2231. The world Ocean Atlas temperature and salinity data is available at https://www.nodc.noaa.gov/OC5/woa13/. The shipboard measurements during cruise M96, M97, and M98 will be available through PANGEA, the DOI is under approval.

### 6 Acknowledgement

We thank Toste Tanhua for closing the M96 section towards the African coast (M97), as well as all the research teams and crews onboard R/V Meteor for their hard work during the three cruise legs. We thank Armin Köhl for providing the GECCO2 data and the information about the data. We also thank Richard Greatbatch for the comments on the manuscript, and Gerd Krahmann for providing the matlab

codes for uCTD sensor calibration. The blended level-4 wind stress data were provided by Copernicus





Marine Environment Monitoring Service (CMEMS). This study is funded by the Deutsche Forschungsgemeinschaft as part of the cooperative project FOR1740 and by the German Federal Ministry of Education and Research as part of the cooperative projects RACE (03F0605B) and SACUS (03F0751A).



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



5   **Table 1: Ekman volume ($M_{Ek}$ in Sv), heat ($H_e$ in PW) and salt ($S_e$ in $10^6$ kg s$^{-1}$) fluxes calculated using different methods, and the transport-weighted temperature $\Theta_E$ and salinity $S_{AE}$ in the Ekman layer. Positive and negative fluxes denote northward and southward fluxes, respectively. The uncertainties of the Ekman heat and salt flux is 0.4 PW and $4.5\times10^6$ kg s$^{-1}$ at 14.5° N, and 0.3 PW and $6.5\times10^6$ kg s$^{-1}$ at 11° S, respectively. The uncertainties of the transport-weighted Ekman temperature and salinity is 0.20 °C and 0.15 g kg$^{-1}$ at 14.5° N, and 0.11 °C and 0.10 g kg$^{-1}$ at 11° S, respectively.**

| Section | | 14.5° N | | | | | 11° S | | | | |
|---|---|---|---|---|---|---|---|---|---|---|---|
| Method | | $\Theta_E$ | $S_{AE}$ | $M_{Ek}$ | $H_e$ | $S_e$ | $\Theta_E$ | $S_{AE}$ | $M_{Ek}$ | $H_e$ | $S_e$ |
| Direct | TTP/profile | 25.52 | 36.33 | 6.21 | 0.413 | 5.40 | 25.41 | 36.83 | -11.71 | -0.842 | -17.69 |
| | TTP/surface | 25.61 | 36.34 | 6.21 | 0.415 | 5.49 | 25.41 | 36.80 | -11.71 | -0.842 | -17.38 |
| Indirect | TTP | 25.46 | 36.32 | 6.68 | 0.443 | 5.72 | 25.13 | 36.81 | -13.64 | -0.965 | -20.50 |
| | Surface | 25.65 | 36.29 | 6.68 | 0.448 | 5.57 | 25.20 | 36.78 | -13.64 | -0.946 | -20.04 |
| | Annual | 26.46 | 36.13 | 8.31 | 0.584 | 5.56 | 25.53 | 36.73 | -11.02 | -0.799 | -15.49 |



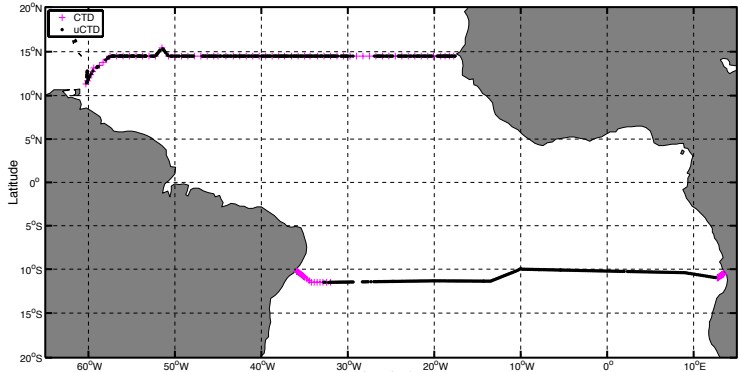

**Figure 1: Positions of the CTD (magenta +) and uCTD (black dots) measurement along 14.5° N and 11° S section. The 14.5° N section was completed during RV Meteor cruises M96 and M97, the 11° S section during M98.**





**Figure 2:** Vertical sections of conservative temperature in °C (a, b), absolute salinity in g kg$^{-1}$ (c, d), neutral density in kg m$^{-3}$ (e, f), geostrophic velocity in cm s$^{-1}$ (g, h), and ageostrophic velocity in cm s$^{-1}$ (i, j) at 14.5° N (left) and 11° S (right). All the available CTD and uCTD data were used to produce (a) to (f), and the contours were plotted with every 5$^{th}$ value for visual clarity. (g) and (i) were calculated using only CTD data, while (h) and (j) were calculated using CTD and uCTD data. The blanks were due to shallow measurement depth of the uCTD.



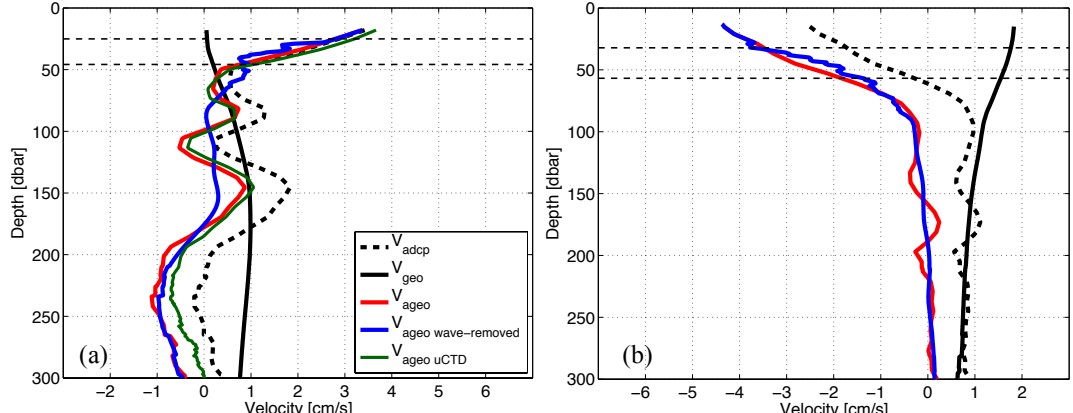

5   **Figure 3: Section-averaged cross-track velocity profiles at (a) 14.5° N and (b) 11° S. The solid red curve is the ageostrophic velocity based on CTD data, the solid black curve is geostrophic velocity based on CTD data, the dashed black curve is ADCP velocity, and the solid blue is the CTD-based ageostrophic velocity after the wave signal being removed (details see text). The dark green curve is the ageostrophic velocity profile based only on uCTD data. The upper horizontal dashed line denotes the basin-wide averaged MLD and the lower denotes the basin-wide averaged TTP depth.**





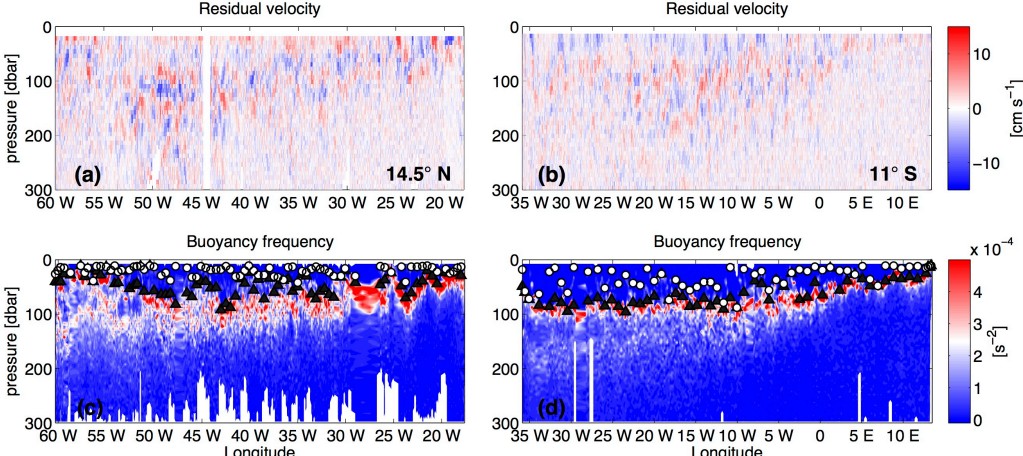

**Figure 4: Vertical sections of residual meridional velocity in cm s⁻¹ at (a) 14.5° N and (b) 11° S and of buoyancy frequency calculated from uCTD/CTD at (c) 14.5° N and (d) 11° S. Northward velocity in (a) and (b) is shaded in red, southward in blue. The residual velocity is calculated by subtracting an 80-m boxcar filtered profile from the original ADCP profile. The white circles in (c) and (d) denote the MLD, the black triangles denotes the TTP (see text for details). The MLD and TTP plotted here are subsampled for visual clarity. No uCTD measurements were conducted between 30° W and 25° W.**



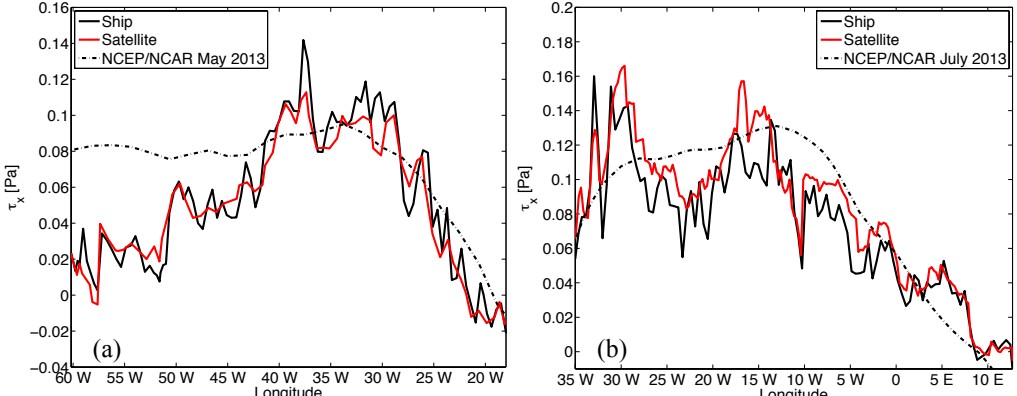

**Figure 5: Zonal wind stress along (a) 14.5° N and (b) 11° S. Ship wind stress (black line) was binned in 50-km interval. The satellite wind stress data (red) were extracted to the nearest ship time and position. The NCEP reanalysis monthly zonal wind stress (black dashed line) at the same latitude in the cruise month is also plotted.**





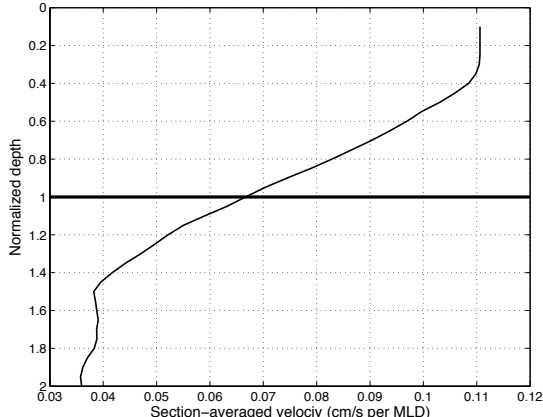

Figure 6: Section-averaged ageostrophic velocity at 14.5° N, normalized in depth by the local MLD. Velocity above 18 m is set equal to the velocity at 18 m. MLD is defined as the depth where the density is 0.01 kg m$^{-3}$ different from the value at 10 m.





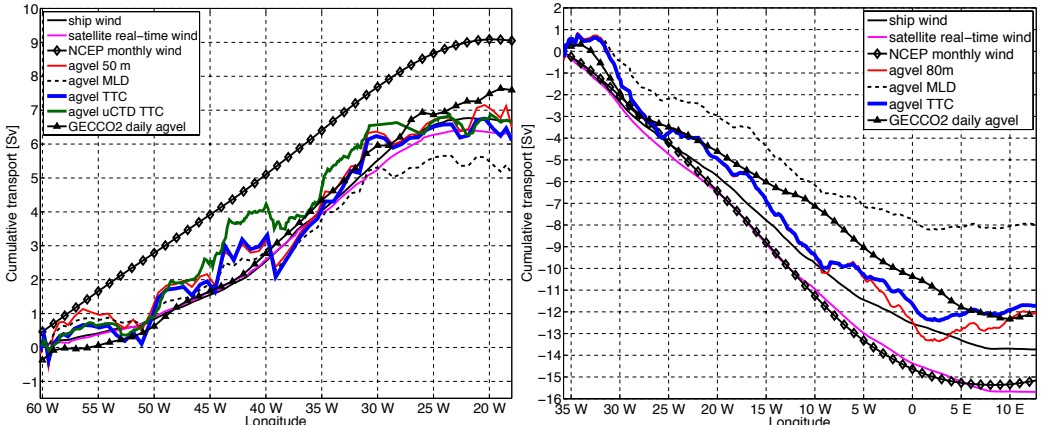

**Figure 7: Cumulative meridional Ekman transport from the western to the eastern coast (a) at 14.5° N and (b) at 11° S. For both sections, the black solid curve marks the indirect Ekman transport estimate from the in-situ wind stress; the magenta solid curve denotes that from the satellite wind stress; and the black diamond line denotes that from the NCEP/NCAR monthly wind stress. The solid blue curve denotes the direct estimate by integrating the ageostrophic velocity to the TTP, the red solid to a uniform depth (50 m at 14.5° N and 80 m at 11° S), the black dashed line to the MLD. The black triangle line represents the direct estimate based on the GECCO2 daily data. The dark green line in (a) represents direct estimate integrated to the TTP based only on the uCTD data.**