# Peer review of "On the meridional ageostrophic transport in the tropical Atlantic"

_Ocean Science, 2016_

## Referee Comment (RC1) · Anonymous Referee #1 · 13 Mar 2017

In this study two cross-basin hydrographic sections closing the Atlantic at 14.5°N and 11°S are used to study the meridional ageostrophic, Ekman and heat and salt transports in the first 300m of the water column. This objective is assessed through hydrographic data (uCTD, CTD, ADCP and TSG), together with in situ winds and wind products (ASCAT, ECMWF and NCEP) and model outputs (GECCO2).

The manuscript is very well written, is well structured, clear, detailed and provides with interesting results of the surface layer. I strongly recommend its publication, after some minor changes.

Mayor changes: Page 19 lines 11-16 and Figure 5. The three wind datasets presents a high agreement between them, with the exception of the westernmost side of 14.5°N, which suggest that the difference in this area cannot be explained by the different drag

coefficient formulation. I believe the source of the difference between winds observed in the western side of the northern transect is linked with some local effect. For example, in other areas, such as the Canary Islands, the QuikScat and NCEP data does not resolve the Von Karman structures that are shed from interactions of the flow with the islands as consequence of a low-resolution topography (e.g. Mason et al. 2011, Pérez-Hernández et al. 2015).

Page 19. How is the uncertainty of the Ekman transport estimated?

Minor changes: Page 3, equation 2. Reading from line 15 to line 19, I understand that equation 2 should be Vageos= Vobs-Vgeos-VageosEkman.

Figure2, Beware that two subplots have been label as (d)

Page 13. In the paragraphs in between lines 6 and 24, could cite Figure 3 or 4 when defining the mean TTP and MLD depths, and same for the respective paragraph on the 11°S section.

References used: Mason, E., F. Colas, J. Molemaker, A. Shchepetkin, C. Troupin, J. McWilliams, and P. Sangr a (2011), Seasonal variability of the Canary Current: A numerical study, J. Geophys. Res., 116, C06001, doi:10.1029/2010JC006665.
 Pérez-Hernández, M. D., G. D. McCarthy, P. Vélez-Belchí, D. A. Smeed, E. Fraile-Nuez, and A. Hernández-Guerra (2015), The Canary Basin contribution to the seasonal cycle of the Atlantic Meridional Overturning Circulation at 268N, J. Geophys. Res. Oceans, 120, doi:10.1002/2015JC010969.

---

## Referee Comment (RC2) · Anonymous Referee #2 · 20 Mar 2017

This manuscript provides estimates of the wind-driven meridional mass, heat, and salt transports across zonal lines at 14.5N and 11S in the tropical Atlantic. The authors use a variety of data sets to estimate the transport, most importantly a collection of measurements from CTD casts and ADCP measurements during cruises across each zonal transect. Because of the difficulty and effort involved with acquiring this kind of data set, and the thorough analysis of the data, these results deserve to be published. They will be useful for others estimating transports from observations and for validating numerical model simulations.

The authors have done a good job describing the methodology and presenting the data. It's good to see that similar results are usually obtained for a variety of methods and wind products (after accounting for deviations of winds during the Lagrangian cruises from monthly mean gridded winds). I have two main suggestions for improvement,

followed by more minor comments and edits.

Main comments:

1. The manuscript contains a lot of description of the data and methodology, and a lot of it is presented in the "Results and discussion" section. For example, the different options for penetration depth of wind-driven currents, level of no motion, methodology for calculating heat and salt transports. I suggest putting most of the data and methodology text into a section (or two) before the results/discussion section. This will improve the flow of the manuscript and allow readers to focus more on the important results instead of being led back and forth between results and methods throughout the manuscript. I also suggest trying to shorten the data/methodology description wherever possible. I give some suggestions in my minor comments below.

2. The discussion of the "bigger picture" can be improved. The authors mention the importance of meridional transport for the AMOC and the connection to the STCs, and there are some comparisons to previous transport estimates, but to a large extent the manuscript presents a detailed set of calculations along two latitude lines during specific times of the year. Error estimates based on aliasing of inertial currents are given, but one also wonders about longer-timescale fluctuations in winds (seasonal, interannual) and how representative the authors' estimates are for annual, seasonal, and monthly climatological mean transport. Some discussion of these considerations would improve the manuscript.

Minor comments:

Section 1: At the end of this section it would be helpful for the authors to describe how their study differs from others (data used, time of year of measurements, etc.).

Section 2: At the beginning of this section (and in figure 1) please indicate the exact dates when each section was occupied.

Section 2.3: Why use two different bulk formulas for wind stress? Can you use the

same for in situ and satellite winds? Same comment for lines 15-16 on p. 19.

Section 3.2: I suggest moving the discussion of reference depth (p. 15, lines 10-25) before description of the geostrophic velocity calculation in the preceding paragraph.

p. 16-18: To focus the manuscript more, I recommend removing, or at least shortening considerably, the parts dealing with the removal/identification of the velocity signal below the TTP since it does not affect the transport calculations.

Section 4: Some discussion of seasonality would be useful. What do you think are the error bars on your estimates, considering seasonal changes in winds and stratification, for example? Or what are the error bars for your weekly/monthly estimates considering interannual variability?

Figure 1: It's difficult to see the CTD locations in the N. Atl. transect. Maybe plot them a little above/below the uCTD marks? Also, maybe add black and white shading of mean zonal wind stress as background and make uCTD marks a different color?

Language edits:

p. 2, line 11: change 'has' to 'have'

p. 2, line 17: change 'they' to 'and'

p. 2, line 20: add hyphen between 'Ekman' and 'driven'

p. 3, lines 3-4: change 'is' to 'was' (two instances)

p. 4, line 1: change 'application' to 'applications' and 'approach' to 'approaches'

p. 4, line 8: delete 'of the direct approach'

p. 7, line 6: delete 'of' before '+/-0.001'

p. 7, line 14: change 'sink' to 'sinks'

p. 7, line 26: change comma to period and begin new sentence with "We..."
p. 7, line 27: change comma to semicolon

p. 7, line 28: change 'allow water passing' to 'allowing water to pass'

p. 8, line 10: change '...leading) and' to '...leading), which'

p. 8, line 20: change 'a' to 'an'

p. 9, line 7: delete 'to'

p. 9, line 15: delete 'the' before 'three'

p. 10, line 1: change 'compare' to 'compared'

p. 11, line 17: insert 'the' after 'from'

p. 12, line 20: change 'isotherm' to 'isotherms'

p. 14, line 14: change 'flew' to 'flowed'

p. 15, line 2: change 'At' to 'Along the'

p. 15, line 7: insert 'and' after 'choice,'

p. 17, lines 26-27: delete comma after 'waves', insert 'a' before 'near-inertial', and insert comma after 'forcing'

p. 18, line 2: delete comma

p. 18, line 20: insert 'and' after comma

p. 18, line 21: change 'decrease' to 'decreases'

p. 18, line 22: change 'appears being' to 'appears to be'

p. 19, line 11: change 'of' to 'in'

p. 19, line 13: insert comma after first 'stress'

p. 19, line 14: change to '...wind speeds from the ship and satellite are very...'

p. 19, line 19: change 'anomalous' to 'anomalously'

p. 19, line 20: change 'were' to 'was'

p. 20, line 5: change 'arises' to 'arise'

p. 20, lines 27-28: delete commas and insert 'of' after 'because'

p. 21, lines 6-7: change to '...motion and therefore...'

p. 21, line 8: insert 'Because' at beginning of sentence

p. 21, line 13: change 'constant as' to 'to equal'

p. 21, line 26: change '...question followed...' to '...question that follows...'

p. 21, line 28: change 'not conclusive' to 'inconclusive'

p. 23, line 10, change 'observation' to 'observations'

p. 23, line 19: insert 'the' after 'that'

p. 23, line 23: change 'estimate based' to 'estimates based on'

p. 23, line 27: change 'fluctuation' to 'fluctuations' and 'this is' to 'which are'

p. 25, line 13: change 'were' to 'was'

p. 26, line 1, change 'estimated' to 'estimate'

p. 27, line 25: change 'is' to 'are'

p. 30, line 12: change 'meridional' to 'zonal'?
* * *

---

## Author Comment (AC1) · 7 May 2017

**Author's response to the referee comments on the manuscript of "On the meridional ageostrophic transport in the tropical Atlantic"**

**Yao Fu, Johannes Karstensen, and Peter Brandt**

yfu@geomar.de

We thank the editor and both referees for the very helpful comments. Here is our response to the comments. It is organized as follows: the referee's comment is first repeated in black colour, followed by the author's response and the corresponding changes in the manuscript in blue colour. At the end, we included a change-tracking version of the manuscript. The author's response part has page number in Roman numerals from I to XIII, the

10 manuscript part has page number in Arabic numerals from 1 to 45.

**Response to the anonymous Referee # 1**

In this study two cross-basin hydrographic sections closing the Atlantic at 14.5N and 11S are used to study the

15 meridional ageostrophic, Ekman and heat and salt transports in the first 300m of the water column. This objective is assessed through hydrographic data (uCTD, CTD, ADCP and TSG), together with in situ winds and wind products (ASCAT, ECMWF and NCEP) and model outputs (GECCO2).

The manuscript is very well written, is well structured, clear, detailed and provides with interesting results of the surface layer. I strongly recommend its publication, after some minor changes.

Thank you very much for the positive evaluation and the following suggestions.

Page 19 lines 11-16 and Figure 5. The three wind datasets presents a high agreement between them, with the exception of the westernmost side of 14.5° N, which suggest that the difference in this area cannot be explained

25 by the different drag coefficient formulation. I believe the source of the difference between winds observed in the western side of the northern transect is linked with some local effect. For example, in other areas, such as the Canary Islands, the QuikScat and NCEP data does not resolve the Von Karman structures that are shed from interactions of the flow with the islands as consequence of a low-resolution topography (e.g. Mason et al. 2011, Pérez-Hernández et al. 2015).

The three wind stress datasets in Figure 5a are ship measured wind stress (black), satellite-based L4 near-real-time wind stress with 6-h temporal resolution (red), and the monthly wind stress from NCEP reanalysis in May 2013 (black dashed). It is expected that the ship and satellite wind stress is very close to each other, since the satellite wind stress was extracted to the nearest ship time and position. However, the biggest difference between the monthly wind stress and the other two in the westernmost part of the 14.5 N section is more likely due to an aliasing effect of the cruise measurement. The temporal variability of the wind is relatively high in the western third of the section. The wind over this region was relatively weak when the cruise started on 28 April 2013, then gradually became stronger during the first week of the cruise. Therefore, the monthly mean wind stress over this region should be stronger than the in-situ wind stress at the measurement time.

At 11 S the satellite wind stress is generally larger than the ship wind stress, while at 14.5 N they tend to agree better. We attributed this to the use of different drag coefficient formulation (COARE 3 for satellite wind product; Large and Yeager, 2004 for ship wind). But we think that the referee's suggestion about the unresolved local effect by the satellite and NCEP data provides another possible explanation to the differences in the different wind stress data. Accordingly, we have added this reason on Page 22, line 23-27.

Page 19. How is the uncertainty of the Ekman transport estimated?

The uncertainty of the indirect Ekman transport is estimated by using the standard deviation of the 6-h NCEP/CFSr wind stress between 2000 and 2011. This is described on page 23, line 8-13. We think that the unresolved temporal variability by a single ship transect is the main source for the uncertainty of the indirect Ekman transport calculation.

Page 3, equation 2. Reading from line 15 to line 19, I understand that equation 2 should be Vageos= Vobs-Vgeos-VageosEkman.

The description on page 3 line 15-17 was not clear enough to correspond with equation 2. Therefore, we modified Eq. 2 and the text as follows:

An ageostrophic velocity ($v_{ageos}$) can be calculated as the difference of the direct observed velocity ($v_{obs}$) and

II

the geostrophic velocity ($v_{geos}$). The ageostrophic velocity might consist of an Ekman component ($v_{Ekman}$) and components that are not in Ekman balance (e.g. inertial currents). Often the non-Ekman components are assumed to be 0, and $v_{Ekman}$ is expected to equal $v_{ageos}$. Under this assumption, the Ekman velocity can be derived as follows:

5    $$v_{Ekman} = v_{obs} - v_{geos}. \quad\quad\quad (2)$$

Now it is on page 3, line 14-19.

Figure2, Beware that two subplots have been label as (d).

10   It is changed.

Page 13. In the paragraphs in between lines 6 and 24, could cite Figure 3 or 4 when defining the mean TTP and MLD depths, and same for the respective paragraph on the 11° S section.

15   Figure 3 and 4 are cited now page 15, between line 3 and line 17.

III

**Response to the anonymous Referee # 2**

This manuscript provides estimates of the wind-driven meridional mass, heat, and salt transports across zonal
lines at 14.5N and 11S in the tropical Atlantic. The authors use a variety of data sets to estimate the transport,
most importantly a collection of measurements from CTD casts and ADCP measurements during cruises across
each zonal transect. Because of the difficulty and effort involved with acquiring this kind of data set, and the
thorough analysis of the data, these results deserve to be published. They will be useful for others estimating
transports from observations and for validating numerical model simulations.

The authors have done a good job describing the methodology and presenting the data.

It's good to see that similar results are usually obtained for a variety of methods and wind products (after
accounting for deviations of winds during the Lagrangian cruises from monthly mean gridded winds). I have two
main suggestions for improvement, followed by more minor comments and edits.

Thank you very much for the positive evaluation and the following suggestions.

Main comments:

1. The manuscript contains a lot of description of the data and methodology, and a lot of it is presented in the
"Results and discussion" section. For example, the different options for penetration depth of wind-driven
currents, level of no motion, methodology for calculating heat and salt transports. I suggest putting most of the
data and methodology text into a section (or two) before the results/discussion section. This will improve the flow
of the manuscript and allow readers to focus more on the important results instead of being led back and forth
between results and methods throughout the manuscript. I also suggest trying to shorten the data/methodology
description wherever possible. I give some suggestions in my minor comments below.

As the referee suggested, we moved some part of the method description from the "Results and discussion"
section to a separate section "Method" before the "Results and discussion" and shortened the descriptions
correspondingly. These include the calculation of the geostrophic and ageostrophic velocity, the error estimation
of the direct Ekman transport calculation, and the different methods for the Ekman heat and salt transport
calculation.

IV

2. The discussion of the "bigger picture" can be improved. The authors mention the importance of meridional transport for the AMOC and the connection to the STCs, and there are some comparisons to previous transport estimates, but to a large extent the manuscript presents a detailed set of calculations along two latitude lines during specific times of the year. Error estimates based on aliasing of inertial currents are given, but one also wonders about longer-timescale fluctuations in winds (seasonal, interannual) and how representative the authors' estimates are for annual, seasonal, and monthly climatological mean transport. Some discussion of these considerations would improve the manuscript.

As the referee suggested, we added a short discussion about the seasonal and interannual variability of Ekman transport, and compared our Ekman transport estimates with an annual climatology calculated from the NCEP/CFSr 6-h wind stress data. It is on page 30, line 19-28.

Minor comments:

Section 1: At the end of this section it would be helpful for the authors to describe how their study differs from others (data used, time of year of measurements, etc.).

There are a number of methodological improvements based on new technologies: we used uCTD data which now offers to do this type of studies from moving ships (e.g. container ships); we used satellite-based wind product and reanalysis wind products, and compared the Ekman transport estimates between these wind products. We also compared the observation-based Ekman transport with the GECCO2 assimilation products at the same location. This is now mentioned at the end of the first section on page 6, line 4-15.

Section 2: At the beginning of this section (and in figure 1) please indicate the exact dates when each section was occupied.

This is added on page 6, line 22-24, page 7, line 1, and in the caption of figure 1 on page 39.

Section 2.3: Why use two different bulk formulas for wind stress? Can you use the same for in situ and satellite winds? Same comment for lines 15-16 on p. 19.

V

The wind speed of the CMEMS product is derived from retrievals of scattermeters ASCAT and OSCAT, and combined with the ECMWF operational wind analysis. The wind stress data is estimated using COARE 3 model (Fairall et al., 2003) and directly available through the CMEMS website. The ship wind stress was calculated by using the bulk formula by Large and Yeager, 2004. In this case, the ship and satellite wind stress agrees very well

5 at 14.5 N, while the ship wind stress is generally weaker than the satellite wind stress at 11 S.

We also calculated the ship wind stress using the COARE 3.0 model. In this case, the ship wind stress tends to agree better with the satellite wind stress at 11 S. However, at 14.5 N, the ship wind stress is generally stronger than the satellite wind stress.

Besides, we also calculated the ship wind stress by using Large and Pond, 1981, Smith, 1988 and Yelland, 1996, and found that the ship wind stress by using Large and Yeager, 2004 returns the indirect Ekman transport estimates that are most consistent with the direct Ekman transport estimates at both sections. If the other bulk formulas were applied to calculate the ship wind stress, the indirect Ekman transport estimates would be much

15 higher than the direct Ekman transport estimates at both sections.

Section 3.2: I suggest moving the discussion of reference depth (p. 15, lines 10-25) before description of the geostrophic velocity calculation in the preceding paragraph.

20 Page 15 line 10-25 describes the calculation of the reference velocity. We re-structured the text following the referee's suggestion. The relative geostrophic velocity calculation from the CTD/uCTD data was first described, than the calculation of the reference velocity from the ADCP data was discussed. Afterwards, the ageostrophic velocity was calculated and the sensitivity of the ageostrophic velocity to the reference depth was discussed. We feel it is necessary to structure the text in this order, since the reference velocity can only be calculated, if the

25 relative geostrophic velocity is first known. This part of the text is now moved to the "Method" section on page 13 and 14.

p. 16-18: To focus the manuscript more, I recommend removing, or at least shortening considerably, the parts dealing with the removal/identification of the velocity signal below the TTP since it does not affect the transport

30 calculations.

VI

Following the referee's suggestion, we shortened the text by deleting the calculation of Ekman transport using a wave-removed ageostrophic velocity and correspondingly modified Fig. 3. Although the identification of the wave-like velocity signal is not conclusive, the information about the horizontal and vertical structure of the ageostrophic velocity and the method of separating this signal from the ADCP velocity might still be interesting
5   for the potential readers. This part of the text is no on page 20-21.

Section 4: Some discussion of seasonality would be useful. What do you think are the error bars on your estimates, considering seasonal changes in winds and stratification, for example? Or what are the error bars for your weekly/monthly estimates considering interannual variability?

The discussion of seasonality is now added on page 30, line 7-17. The error bars for the monthly estimates at 14.5 N and 11 S is now given by the standard deviation of the monthly mean Ekman transport in May and July between 1979 and 2010 by using the NCEP/NCAR monthly wind stress. In this way, the interannual variability of the monthly Ekman transport estimates at the two latitudes in the respective month should be taken into
15   account. It is now on page 23, line 11-13.

Figure 1: It's difficult to see the CTD locations in the N. Atl. transect. Maybe plot them a little above/below the uCTD marks? Also, maybe add black and white shading of mean zonal wind stress as background and make uCTD marks a different colour?

In Figure 1, the uCTD position at 14.5 N is now shifted 0.5 to the north. A mean zonal wind stress calculated from NCEP/CFSr monthly wind stress between 1979 and 2011 is plotted as grey background shading with contours. The uCTD position is now marked as blue dots. The caption of Figure 1 is modified correspondingly.

25   Language edits:
    p. 2, line 11: change 'has' to 'have'

    Changed.

30   p. 2, line 17: change 'they' to 'and'

Changed.

p. 2, line 20: add hyphen between 'Ekman' and 'driven'

Added.

p. 3, lines 3-4: change 'is' to 'was' (two instances)

Changed.

p. 4, line 1: change 'application' to 'applications' and 'approach' to 'approaches'

Changed, now on page 4, line 3-4.

p. 4, line 8: delete 'of the direct approach'

Deleted.

p. 7, line 6: delete 'of' before '+/-0.001'

Deleted.

p. 7, line 14: change 'sink' to 'sinks'

Changed, now on page 7, line 19.

p. 7, line 26: change comma to period and begin new sentence with "We..."

Changed, now on page 8, line 5.

p. 7, line 27: change comma to semicolon

VIII

p. 7, line 28: change 'allow water passing' to 'allowing water to pass'

p. 8, line 10: change '...leading) and' to '...leading), which'

p. 8, line 20: change 'a' to 'an'

p. 9, line 7: delete 'to'

Deleted.

20 p. 9, line 15: delete 'the' before 'three'

Deleted.

p. 10, line 1: change 'compare' to 'compared'

p. 11, line 17: insert 'the' after 'from'

IX

p. 12, line 20: change 'isotherm' to 'isotherms'

Changed, now on page 19, line 3.

5  p. 14, line 14: change 'flew' to 'flowed'

Changed, now on page 20, line 2.

p. 15, line 2: change 'At' to 'Along the'

Changed, now on page 13, line 12.

p. 15, line 7: insert 'and' after 'choice,'

15  Inserted, now on page 13, line 17.

p. 17, lines 26-27: delete comma after 'waves', insert 'a' before 'near-inertial', and insert comma after 'forcing'

Done, now on page 21, line 20-21.

p. 18, line 2: delete comma

Deleted.

25  p. 18, line 20: insert 'and' after comma

This part of the text is removed.

p. 18, line 21: change 'decrease' to 'decreases'

This part of the text is removed.

X

p. 18, line 22: change 'appears being' to 'appears to be'

This part of the text is removed.

p. 19, line 11: change 'of' to 'in'

Changed, now on page 22, line 15.

10   p. 19, line 13: insert comma after first 'stress'

Inserted, now on page 22, line 17.

p. 19, line 14: change to '...wind speeds from the ship and satellite are very...'

Changed, now on page 22, line 18.

p. 19, line 19: change 'anomalous' to 'anomalously'

20   Changed, now on page 22, line 22.

p. 19, line 20: change 'were' to 'was'

Changed, now on page 22, line 23.

p. 20, line 5: change 'arises' to 'arise'

Changed, now on page 23, line 13.

30   p. 20, lines 27-28: delete commas and insert 'of' after 'because'

XI

p. 21, lines 6-7: change to '...motion and therefore...'

p. 21, line 8: insert 'Because' at beginning of sentence

p. 21, line 13: change 'constant as' to 'to equal'

15  p. 21, line 26: change '...question followed...' to '...question that follows...'

p. 21, line 28: change 'not conclusive' to 'inconclusive'

p. 23, line 10, change 'observation' to 'observations'

p. 23, line 19: insert 'the' after 'that'

p. 23, line 23: change 'estimate based' to 'estimates based on'

XII

p. 23, line 27: change 'fluctuation' to 'fluctuations' and 'this is' to 'which are'

p. 25, line 13: change 'were' to 'was'

p. 26, line 1, change 'estimated' to 'estimate'

p. 27, line 25: change 'is' to 'are'

20 p. 30, line 12: change 'meridional' to 'zonal'?

XIII

[revised manuscript text omitted]